# FLIP: Flow-Centric Generative Planning as General-Purpose Manipulation World Model

**Chongkai Gao**
National University of Singapore
gaochongkai@u.nus.edu

**Haozhuo Zhang**
Peking University
2100013132@stu.pku.edu.cn

**Zhixuan Xu, Zhehao Cai**
National University of Singapore
{zhixuanxu,e1373791}@u.nus.edu

**Lin Shao**
National University of Singapore
linshao@nus.edu.sg

## ABSTRACT

We aim to develop a model-based planning framework for world models that can be scaled with increasing model and data budgets for general-purpose manipulation tasks with only language and vision inputs. To this end, we present FLow-CentrIc generative Planning (FLIP), a model-based planning algorithm on visual space that features three key modules: 1) a multi-modal flow generation model as the general-purpose action proposal module; 2) a flow-conditioned video generation model as the dynamics module; and 3) a vision-language representation learning model as the value module. Given an initial image and language instruction as the goal, FLIP can progressively search for long-horizon flow and video plans that maximize the discounted return to accomplish the task. FLIP is able to synthesize long-horizon plans across objects, robots, and tasks with image flows as the general action representation, and the dense flow information also provides rich guidance for long-horizon video generation. In addition, the synthesized flow and video plans can guide the training of low-level control policies for robot execution. Experiments on diverse benchmarks demonstrate that FLIP can improve both the success rates and quality of long-horizon video plan synthesis and has the interactive world model property, opening up wider applications for future works. Video demos are on our website: https://nus-lins-lab.github.io/flipweb/.

## 1 INTRODUCTION

World models refer to learning-based representations or models that learn to simulate the environment (LeCun, 2024; Ha & Schmidhuber, 2018). With world models, agents can imagine, reason, and plan inside world models to solve tasks more safely and efficiently. Recent advancements in generative models, especially in the area of video generation (Brooks et al., 2024; Blattmann et al., 2023; Yang et al., 2023), have demonstrated the application of generating high-quality videos as world simulators with internet-scale training data. World models have opened new avenues across various fields, particularly in the domain of robotic manipulation (Yang et al., 2023; Mendonca et al., 2023; Seo et al., 2023), which is the focus of this paper.

The intelligence of generalist robots involves two levels of abilities (Caucheteux & King, 2022; Manto et al., 2012): 1) high-level planning of the abstraction sequence of the task with multi-modal inputs, and 2) low-level execution of the plan by interacting with the real world. A well-designed world model could serve as an ideal way to realize the first function, for which it should enable model-based planning. This requires the world model to be interactive, i.e., can simulate the world according to some given actions. The core of this framework is to find a *scalable action representation* that serves as the connection between high-level planning and low-level control. This representation should: 1) be able to represent various kinds of movements across diverse objects, robots, and tasks in the whole scene; 2) be easy to obtain or label a large amount of training data for scaling up. Regarding this, Yang et al. (2023); Du et al. (2023); Zhou et al. (2024) use languages from VLMs (Driess et al., 2023) as high-level actions, while Wu et al. (2024) directly use low-level robot actions to interact with the world model. However, they either require extra datasets or task-specific high-level action labeling processes for training the interactive world model, or their representations cannot describe sophisticated subtle movements in the whole scene. For example, they cannot describe the detailed movements of a dexterous hand spinning a pen. These limit their application as a scalable interactive world model and inspire us to find other action representations.

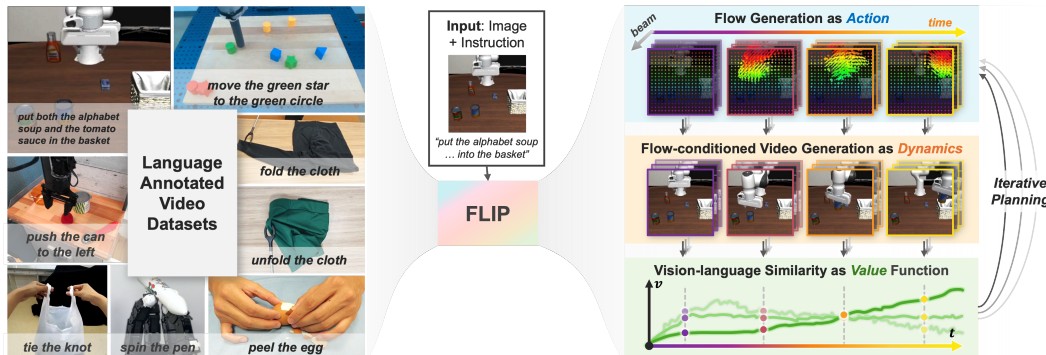

Figure 1: Overview of our method. Left: FLIP is trained on video datasets across different tasks, objects, and robots, with only one language description for each video as the goal. Right: we train an interactive world model consisting of an action module for flow generation, a dynamics module for video generation, and a value module for assigning value at each step. These modules can perform flow-centric model-based planning for manipulation tasks on the flow and video space.

Image flow, a dynamic representation of pixel-level changes over time, is a concise yet general representation of all kinds of movements in images for different robots and objects and can describe more subtle changes than language. More importantly, image flow can be completely obtained by off-the-shelf trackers (Karaev et al., 2023) from pure video datasets. Meanwhile, recent works also show that flows are effective representations for training low-level manipulation policies (Wen et al., 2023; Xu et al., 2024a;b). These make flow a good choice for action representation of world models. However, it remains unclear how to leverage flows for planning on manipulation tasks.

In this work, we present Flow-Centric General Planning (FLIP) for general-purpose robot manipulation tasks. As shown in Figure 1, we train a flow-centric world model purely from language-annotated video data from diverse tasks. This world model contains three modules: 1) a flow generation network as the action module, 2) a flow-conditioned video generation model as the dynamics module, and 3) a visual-language representation learning model as the value module. Specifically, we design our action and dynamics module based on CVAE (Kingma, 2013) and DiT (Peebles & Xie, 2023) architectures respectively and propose a new training mechanism for leveraging LIV (Ma et al., 2023) as our value module. The trained modules enable model-based planning by progressively searching successful long-horizon plans on the flow and video spaces: given an initial image and language instruction as the goal, the action module will propose several flow candidates, and the dynamics module will generate the short-horizon future videos. The value module will access the favorability of generated videos that maximize the discounted returns and perform tree search (Selman & Gomes, 2006) to synthesize long-horizon plans for solving the task.

In experiments, we show that FLIP can perform model-based planning to solve tasks for both simulation manipulation tasks (LIBERO (Liu et al., 2024a)) and real-world tasks (including FMB (Luo et al., 2023), cloth folding, unfolding, and Bridge-V2 (Walke et al., 2023)). We also show that FLIP can generate high-quality long-horizon videos for these tasks. Meanwhile, the generated flow and video plans can guide the training of low-level policies. We also show that the three modules of FLIP are superior to their respective baselines (Wen et al., 2023; Zhu et al., 2024; Ma et al., 2023). We quantitatively show that FLIP can simulate diverse complex manipulation tasks across objects and robots. The trained world model also demonstrates interactive properties, zero-shot transfer ability, and scalability. In summary, our contributions are:

- We propose flow-centric generative planning (FLIP) as an interactive world model for general-purpose model-based planning for manipulation tasks.

- We design a new flow generation network, a new flow-conditioned video generation network, and a new training method for an existing vision-language representation learning network as the three key modules of FLIP.

- In our experiments, we show FLIP can perform general-purpose model-based planning, synthesize long-horizon videos, guide the training of low-level policy, and other promising properties, as well as the superiority of the three modules of FLIP compared to baselines.

## 2 RELATED WORKS

### 2.1 WORLD MODELS FOR DECISION MAKING

Early works of world models learn system dynamics in low dimensional state space (Lesort et al., 2018; Ferns et al., 2004), perform planning in latent space (Nasiriany et al., 2019), or train networks to predict the future observations (Finn & Levine, 2017) and actions (Kaiser et al., 2019). Modern model-based reinforcement learning methods (Hafner et al., 2020; 2023; Hansen et al., 2023; Baker et al., 2022; Micheli et al., 2022) focus on latent space imagination with coupled dynamics and action modules. Recent works leverage powerful scalable video generation architectures like Diffusion Transformer (Peebles & Xie, 2023) and large-scale training data (Grauman et al., 2022) to develop video generation networks to simulate an interactive environment (Yang et al., 2023; Bruce et al., 2024; Shridhar et al., 2024; Valevski et al., 2024; Wu et al., 2024; Zhu et al., 2024; Wu et al., 2024). In this work, we build a world model with separate flow-centric action and dynamics modules as well as a vision-language value model for model-based planning for robot manipulation tasks.

### 2.2 FLOW AND VIDEO MODELS FOR MANIPULATION

Flows are the future trajectories of query points on images or point clouds. They are universal descriptors for motions in the video, while video data contains rich knowledge of behaviors, physics, and semantics, and have unparalleled scalability in terms of both content diversity and data acquisition. For robotics, people have been trying to use flows as policy guidance (Wen et al., 2023; Bharadhwaj et al., 2024), learn dense correspondence (Jiang et al., 2024b), tool using (Seita et al., 2023), or cross-embodiment representations (Xu et al., 2024a; Zhu et al., 2024; Yuan et al., 2024). Videos are usually used for learning inverse dynamics (Du et al., 2024; Finn & Levine, 2017; Brandfonbrener et al., 2024; Gao et al., 2021), rewards (Ma et al., 2022; 2023; Nair et al., 2022; Zakka et al., 2022), transferrable visual representations such as latent embeddings (Sermanet et al., 2018; Nair et al., 2022; Liu et al., 2024a), key points (Huang et al., 2024; Di Palo & Johns, 2024), affordance (Bahl et al., 2023; Shu et al., 2017), flows (Wen et al., 2023; Xu et al., 2024a; Bharadhwaj et al., 2024), scene graphs (Zhang et al., 2024; Jiang et al., 2024a; Kumar et al., 2023), or acquire similar manipulation knowledge from human videos (Wang et al., 2023; Mendonca et al., 2023; Shao et al., 2021; Liang et al., 2024). Recent works also use video generation techniques as visual simulation (Yang et al., 2023; Liu et al., 2024b). In this work, we build our action, dynamics, and value modules all based on video and language inputs, enabling the scalability of our framework.

## 3 THREE FUNDAMENTAL MODULES OF FLIP

### 3.1 PROBLEM FORMULATION

We model a manipulation task $\mathcal{T}$ as a goal-conditioned Partially Observable Markov Decision Process (POMDP) parameterized by $(\mathcal{S}, \mathcal{O}, \phi, \mathcal{A}, P, R, \gamma, g)$ where $\mathcal{S}, \mathcal{A}, \mathcal{O}$ are state, action, and observation spaces, $\phi : \mathcal{S} \rightarrow \mathcal{O}$ is the state-observation mapping function, $P : \mathcal{S} \times \mathcal{A} \rightarrow \mathcal{S}$ is the transition function, $R : \mathcal{S} \times \mathcal{A} \rightarrow \mathbb{R}$ is the reward function, $\gamma$ is the discount factor, and $g$ is the goal state. In this work, the observation space is the image space: $\mathcal{O} = \mathbb{R}^{H \times W \times 3}$, where $H$ and $W$ are the height and width of the image, and $R(s, g) = \mathbb{I}(s == g) - 1$ is a goal-conditioned sparse reward. The task is solved if the agent maximizes the return $\sum_{t=0}^{T} \gamma^t R(s_t, g)$.

We aim to solve this problem by learning a world model and a low-level policy. The world model performs model-based planning on image and flow spaces to maximize the return, synthesizing long-horizon plans, and the low-level policy executes the plan in the real environment. We aim to train the world model only on language-annotated video datasets to make it general-purpose and scalable, and train the low-level policy on a few action-labeled datasets. To enable model-based planning, our world model contains three key modules, as introduced in the following sections.

### 3.2 FLOW GENERATION NETWORK AS ACTION MODULE

**Overview.** The action module of FLIP is a flow generation network $\pi_f$ that generates image flows (future trajectories on query points) as *actions* for planning. The reason why we use a generation model rather than a predictive model is that we are doing model-based planning, where the action module should give different action proposals for sampling-based planning. Formally, given $h$ step

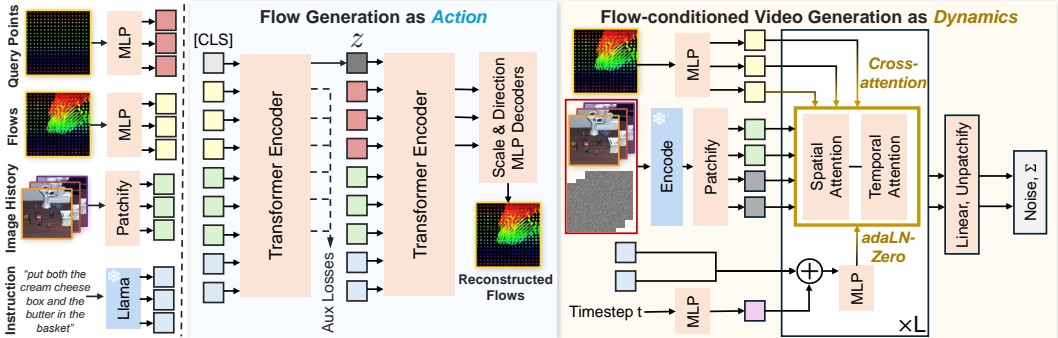

Figure 2: The action module and dynamics module of FLIP. Left: the tokenizing process of different modalities in training data. Middle: we use a Conditional VAE to generate flows as actions. It separately generates the delta scale and directions on each query point for flow reconstruction. Right: we use a DiT model with the spatial-temporal attention mechanism for flow-conditioned video generation. Flows (and observation history) are conditioned with cross attention, while languages and timestep are conditioned with AdaLN-zero.

image observation history $o_{t-h:t}$ at timestep $t$, the language goal $g$, and a set of 2D query points coordinates $\mathbf{p}_t = \{p_t^k\}_{k=1}^K$, where $p_{t,k} = (u, v)$ is the $k$-th query point coordinate on $o_t$, the flow generation network $\pi_f$ generates coordinates of query points in future $L$ timesteps (including the current step): $\mathbf{p}_{t:t+L} = \pi_f(o_{t-h:t}, \mathbf{p}_t, g) \in \mathbb{R}^{L \times K \times 2}$.

**Training Data Annotation.** The flows of query points can be extracted from pure video data by the off-the-shelf point tracking models. The problem is how to select query points. Previous works either use SAM (Ravi et al., 2024) to select query points on the region of interest or select query points on active/stationary regions with a predefined ratio (Wen et al., 2023). These methods face two problems: 1) for diverse kinds of videos with complex scenes, it is hard for modern segmentation models (Ravi et al., 2024) to segment perfect regions of interest with no human assistance; 2) for long-horizon videos, there may be objects appearing/disappearing in the video, and using query points only from the initial frame become problematic. To this end, in this work, we uniformly sample dense grid query points for the whole image (for the first problem) at each timestep, and track them for only a short-horizon video clip, i.e., tracking on video clips starting from *every* frame of the long-horizon video (for the second problem). This can mitigate the second problem because even if some objects appear/disappear, their influences are restricted in a short horizon. Formally, for each frame in the dataset, we uniformly sample a grid of $N_q$ points, then use Co-Tracker (Karaev et al., 2023) to generate the flows $\{p_{t:t+L}^k\}_{k=1}^{N_q}$ within a future video clip of $L$ steps.

**Model Design.** We design a Conditional VAE (Kingma, 2013) with transformer (Vaswani, 2017) architecture for flow generation, as illustrated in Figure 2. As opposed to previous flow prediction works (Wen et al., 2023; Xu et al., 2024a; Bharadhwaj et al., 2024), we observe enhanced performance when predicting relative displacements rather than absolute coordinates, i.e., we predict $\Delta p_t^k = p_{t+1}^k - p_t^k$ for the $k$-th point at each time step.

For the VAE encoder, we encode ground truth flow $\{\mathbf{p}_t\}_{t=1}^L$, patchify observation history $o_{t-h:t}$, and encode language embedding from Llama 3.1 8B (Dubey et al., 2024) to tokens, concatenate them with a *CLS* token for gathering the information, and then send them to a transformer encoder to extract the output at the *CLS* token position as the latent variable of VAE. For the VAE decoder, we first encode the query points $\{p_t^k\}_{k=1}^{N_q}$ at only timestep $t$ to query tokens, concatenate them with image and language tokens as well as the sampled latent variable $z$ from reparameterization, and send them to another transformer encoder. We extract the output at the query tokens and use two MLPs to predict the delta scale $\delta_s \in \mathbb{R}_{\geq 0}^{L \times K}$ and delta direction $\vec{\delta_d} \in \mathbb{R}^{2L \times K}$ for $L$ future horizons.

Thus we can get $\Delta p_t^k = \delta_s^{tk} \vec{\delta_d^{tk}}$, and the whole future flow can be reconstructed step by step. We also decode the output at the image token positions as an auxiliary image reconstruction task (Wen et al., 2023; He et al., 2022a), which we find useful for improving the training accuracy.

### 3.3 FLOW-CONDITIONED VIDEO GENERATION NETWORK AS DYNAMICS MODULE

**Overview.** The flow-conditioned video generation network $\mathcal{D}$ generates the following $L$ frames based on current image observation history $o_{t-h:t}$, the language goal $g$, and the predicted flow $\mathbf{p}_{t:t+L}$ to enable iterative planning for the next planning step: $\hat{o}_{t+1:t+L} = \mathcal{D}(o_{t-h:t}, g, \mathbf{p}_{t:t+L})$.

**Model Design.** We design a new latent video diffusion model that can effectively take as input different kinds of conditions such as images, flows, and language. This model is built on the DiT (Peebles & Xie, 2023) architecture with spatial-temporal attention mechanism (Ma et al., 2024; Bruce et al., 2024; Zhu et al., 2024). The background knowledge of latent video diffusion models is in Appendix A.1. Here we introduce the design of the multi-modal condition mechanism.

In the original DiT (Peebles & Xie, 2023) and previous trajectory-conditional video diffusion paper (Zhu et al., 2024), they use adaptive layer norm zero (AdaLN-Zero) blocks to process conditional inputs (such as diffusion timestep and class labels), which regress the scale and shift parameters of the layer norm layers from all conditions with a zero-initialized MLP. However, AdaLN will compress all conditional information to scalars, and cannot enable fine-grained interaction between different parts of conditions with the inputs. Thus, this mechanism is not suitable for complex conditions such as image and flow (Zhang et al., 2023; Bao et al., 2023). To this end, we propose a mixed conditioning mechanism for multi-modal conditional generation. We use cross attention for fine-grained interactions between flow conditions (tokenized as $N_q$ tokens) and observation conditions and noisy frames. For image history conditions, we concatenate them on the Gaussian noise frames. We use AdaLN-Zero to process the global conditions including the diffusion timestep and language instruction, as shown in Figure 2. To keep the observation condition clean, we do not add noise to $o_{t-h:t}$ during the diffusion process and do not perform denoising on them either.

### 3.4 VISION-LANGUAGE REPRESENTATION LEARNING AS VALUE MODULE

**Overview.** The value module $\mathcal{V}$ assigns an estimated value $\hat{V}_t$ for each frame $o_t$ to enable model-based planning on the image space, based on the language goal $g$: $\hat{V}_t = \mathcal{V}(o_t, g)$. In this work, we adopt LIV (Ma et al., 2023) to instantiate the value function. LIV first learns a shared language-vision representation from action-free videos with language annotations. It then computes the similarity between current frame $o_t$ and $g$ as the value for timestep $t$: $\hat{V}_t = \mathcal{S}(\psi_I(o_t), \psi_L(g)) = \frac{1}{1-\gamma} cos(\psi_I(o_t), \psi_L(g))$, where $\psi_I$ and $\psi_L$ are the encoding network for image and language respectively, and $\mathcal{S}$ is the $\gamma$-weighted cosine similarity metric.

The pretrained LIV model needs to be fine-tuned to give good value representation on new tasks (Ma et al., 2023). The original fine-tuning loss $\mathcal{L}_{LIV} = \mathcal{L}_I(\psi_I) + \mathcal{L}_L(\psi_I, \psi_L)$ is calculated on sampled sub-trajectory batch data

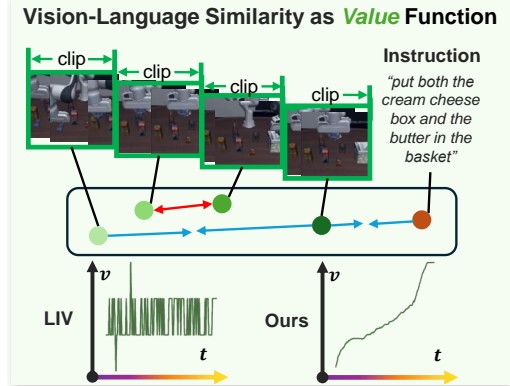

Figure 3: Top: The value module of FLIP. We follow the idea of (Ma et al., 2023) and use time-contrastive learning for the visual-language representation, but we treat each video clip (rather than each frame) as a state. Bottom: the fine-tuned value curves of Ma et al. (2023) and ours.

$\left\{o_s^i, \ldots, o_k^i, o_{k+1}^i, \ldots, o_T^i, g^i\right\}_{i=1}^B$ from each task $\mathcal{T}_i$, where $s \in [0, T_i - 1], s \leq k < T_i$. For $\forall$ task $i$, $\mathcal{L}_I(\psi_I)$ will use time contrastive learning to increase the similarity $\mathcal{S}(o_s^i, o_T^i)$ between the sampled start frame and the end frame and keep the embedding distance between two adjacent frames as ($\gamma$-discounted) 1, and $\mathcal{L}_L$ encourages the image goal $o_T^i$ and language goal $g^i$ have the same embedding for the same task $i$. Details of this process can be found in Appendix A.2.

**Finetuning LIV on Long-Horizon Imperfect Videos.** Finetuning LIV with the original training objective works well on short-horizon perfect videos (about 50 frames in their original papers (Ma et al., 2022; 2023)). However, we find that it does not work well for our long-horizon imperfect videos, as shown in Figure 3, where the fine-tuned value curve exhibits numerous jagged peaks.

---

**Algorithm 1** Flow-Centric Generative Planning

---

1: **Input:** Current observation history $o_{0-h:0}$, language goal $g$, query points $\mathbf{p}$, flow prediction network $\pi_f$, flow-conditioned video generation network $\mathcal{D}$, vision-language value module $\mathcal{V}$.
2: **Hyperparameters:** Flow candidates number $A$, planning Beams $B$, planning horizon $H$.
3: **Initialization:** Flow plans $f_p \leftarrow [\mathbf{p}^i_{0-h:0}, i \in 1 \dots B]$, video plans $v_p \leftarrow [o^i_{0-h:0}, i \in 1 \dots B]$.
4: **for** $h = 1 \dots H$ **do**
5:     **for** $b = 1 \dots B$ **do**
6:         $o \leftarrow v_p[b][-h :]$         ▷ Get the Latest Observation History in the Plan Beam
7:         $a_{1:A} \leftarrow \pi(o, \mathbf{P}, g)$         ▷ Generate A Different Flow Actions
8:         $v_{1:A} \leftarrow \mathcal{D}(o, a_i, g)$ for $i$ in $(1 \dots A)$     ▷ Generate Corresponding Different Video Clips
9:         $id \leftarrow \arg \max \mathcal{R}(v_i, g)$ for $i$ in $(1 \dots A)$
10:        $f_p[b].\text{append}(a_{id}), v_p[b].\text{append}(v_{id})$         ▷ Add Plans with Highest Value
11:     **end for**
12:     $max\_idx, min\_idx \leftarrow \arg \max(v_p, \mathcal{V}), \arg \min(v_p, \mathcal{V})$
13:     $f_p[min\_idx] \leftarrow f_p[max\_idx], v_p[min\_idx] \leftarrow v_p[max\_idx]$     ▷ Periodically Replace
14: **end for**
15: $f \leftarrow f_p[\arg \max(v_p, \mathcal{V})], v \leftarrow v_p[\arg \max(v_p, \mathcal{V})]$     ▷ Return Highest Value Plan

---

This is disastrous for sampling-based planning algorithms since most planning algorithms expect a smoothing value curve to be effective (Selman & Gomes, 2006; Ma et al., 2023).

We point out that this problem is caused by imperfect long-horizon videos, where the task does not necessarily progress smoothly as the video progresses. For example, the robot arm may hesitate in the air during the task. To mitigate this problem, we replace the concept of *adjacent frames* in the original loss to *adjacent states*, where we define states as short-horizon video clips. Formally, we divide a long-horizon video into small segments of fixed length and treat each clip $s^{clip}$ as the smallest unit of the video. The original $o_s, o_T, o_k, o_{k+1}$ are seamlessly replaced by $s^{clip}_s, s^{clip}_T, s^{clip}_k, s^{clip}_{k+1}$ respectively. As shown in Figure 3, this simple strategy is surprisingly useful and makes the fine-tuned vale curve much smoother than the originally fine-tuned ones.

## 4 FLOW-CENTRIC GENERATIVE PLANNING

### 4.1 MODEL-BASED PLANNING WITH FLOWS, VIDEOS, AND VALUE FUNCTIONS

Directly generating long-horizon videos autoregressively is usually not accurate (Wen et al., 2023; Yang et al., 2023; Du et al., 2024) due to compounding errors. In this work, we use model-based planning to search for a sequence of flow actions and video plans that maximizes the discounted return:

$$o^*_{0:L} = \arg \max_{o_{0:L} \sim \pi_f, \mathcal{D}} \sum_{i=0}^{L} \gamma^i R(o_i, g). \tag{1}$$

According to Bellman Equation (Sutton, 2018), this equals stepping towards the next state that maximizes $r_t + \gamma V^*(s_{t+1}, g)$ at each time step given an optimal value function $V^*$. In our problem, $r_t = -1$ is a constant for every step before reaching the goal, and we assume our learned value function $\mathcal{V} = V^*$, thus our problem is simplified to find the next state that maximizes $\mathcal{V}$ at each time step. Note this reward design also encourages finding the shortest plan. We use hill climbing (Selman & Gomes, 2006) to solve this problem. It initializes $B$ plan beams. At each timestep $t$, given current image history $o_{t-h:t}$ and the language goal $g$, it employs $\pi_f$ to generate multiple flow actions $\mathbf{p}_{t+1:t+L} = \pi_f(o_{t-h:t}, \mathbf{p}_t, g)$ on uniformly sampled query points as candidates for tree search, then use $\mathcal{D}$ to generate corresponding short-horizon videos $o_{t+1:t+L} = \mathcal{D}(o_{t-h:t}, g, \mathbf{p}_{t+1:t+L})$. The value module $\mathcal{V}$ is then used to select the generated video with the highest reward among $A$ videos to enable the next iteration of generation for each beam. In order to prevent exploitative planning routes that over-exploit on an irregular state, we periodically replace the lowest value plan among the beams with the beam with the highest value. The algorithm is summarized in Algorithm 1.

| | LIBERO-LONG (Liu et al., 2024a) | FMB-S (Luo et al., 2023) | FMB-M | Folding | Unfolding |
|---|---|---|---|---|---|
| UniPi (Du et al., 2024) | 2% | 0% | 0% | 20% | 10% |
| FLIP-NV | 78% | 52% | 40% | **100%** | 70% |
| FLIP(Ours) | **100%** | **86%** | **78%** | **100%** | **90%** |

Table 1: Success rates of model-based planning on long-horizon tasks.

| | LIBERO-LONG (Liu et al., 2024a) | | | FMB (Luo et al., 2023) | | | Bridge-V2 (Walke et al., 2023) | | |
|---|---|---|---|---|---|---|---|---|---|
| | Latent L2 ↓ | FVD ↓ | PSNR ↑ | Latent L2 ↓ | FVD ↓ | PSNR ↑ | Latent L2 ↓ | FVD ↓ | PSNR ↑ |
| LVDM He et al. (2022b) | 0.566 | 610.98 | 10.852 | 0.484 | 358.22 | 12.349 | 0.373 | 153.41 | 16.481 |
| IRASim (Zhu et al., 2024) | 0.407 | 206.28 | 12.205 | 0.395 | 172.45 | 13.157 | 0.325 | 138.97 | 16.796 |
| FLIP(Ours) | **0.217** | **35.62** | **26.452** | **0.264** | **43.712** | **25.531** | **0.173** | **36.15** | **33.485** |

Table 2: Quantitative results on long-horizon video generation.

## 4.2 PLAN-CONDITIONED LOW-LEVEL POLICY

The low-level policy $\pi_L$ are given the image observation history $o_{t-h:t}$, the language goal $g$, and the predicted flow plan $\mathbf{p}_{t:t+L}$ as well as the video plan $o_{t+1:t+L} = \mathcal{D}(o_{t-h:t}, g, \mathbf{p}_{t+1:t+L})$ to predict the low-level robot action $a_{t:t+L}$ that drive the robot to operate in the environment. We train different policies that take as input different kinds of condition information, with all of them trained on a few demonstrations with action labels. The policy architectures are similar to diffusion policy (Chi et al., 2023). Details can be found in Appendix A.3.

## 5 EXPERIMENTS

In this section, we first demonstrate that FLIP can: 1) perform model-based planning for different manipulation tasks; 2) synthesize long-horizon videos ($\geq$ 200 frames); and 3) can guide the low-level policy for executing the plan for both simulation and real-world tasks. We also evaluate the action, dynamics, and value modules separately compared to corresponding baselines and show the interactive, zero-shot, scalability properties of FLIP. More results and videos are on our website.

### 5.1 MODEL-BASED PLANNING FOR MANIPULATION TASKS

**Setup.** In this section, we train FLIP on four benchmarks to show its model-based planning ability. The model is given an initial image and a language instruction, and it is required to search the flow and video spaces to synthesize the plan for this task. The first one is LIBERO-LONG (Liu et al., 2024a), a long-horizon table-top manipulation benchmark of 10 tasks in simulation. We train FLIP on $50 \times 10$ long-horizon videos with a resolution of $128 \times 128 \times 3$ and test on $50 \times 10$ new random initializations. The second one is the FMB benchmark (Luo et al., 2023), a long-horizon object manipulation and assembly benchmark with varying object shapes and appearances. We train FLIP on 1K single-object multi-stage videos and 100 multi-object multi-stage videos with a resolution of $128 \times 128 \times 3$ and test on 50 new initialization for each. The third and fourth suites are cloth folding and cloth unfolding. These two datasets are collected by ourselves. We train each suite on 40 videos with varying viewpoints and test on 10 new viewpoints for each with a resolution of $96 \times 128 \times 3$.

We follow previous works(Du et al., 2023; Zhu et al., 2024) and evaluate our model-based planning results by human evaluating the correctness of generated video plans. That is, we visually assess the percentage of time the video successfully solved the given task. We compare FLIP to two baselines: 1) UniPi (Du et al., 2024), a text-to-video generation method with long-horizon text goals. 2) FLIP-NV, an ablation of FLIP that performs the same beam search but with no value module as guidance.

**Results.** Table 1 shows the results. We can see that UniPi achieves low success rates across all tasks, which shows that directly synthesizing long-horizon videos is difficult. FLIP-NV achieves better results than UniPi. This shows that with dense flow information as guidance, the performance of the video generation model is improved. FLIP outperforms all baselines, pointing out the effectiveness of using value functions for model-based planning. This can eliminate incorrect search routes during planning. We show such incorrect search routes on our website.

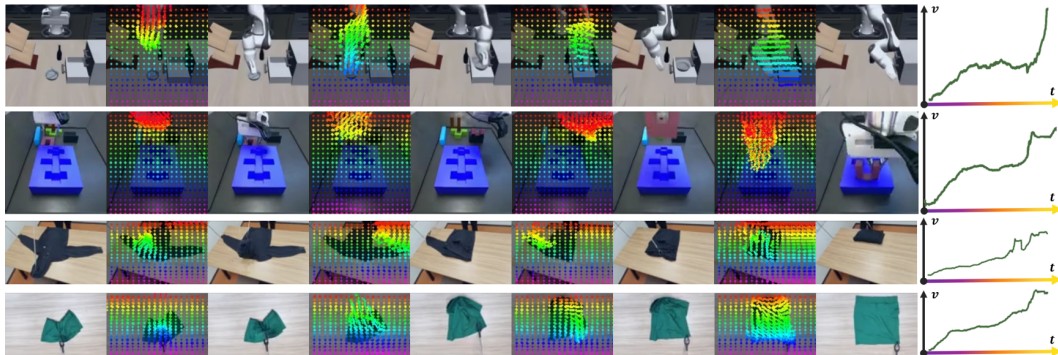

Figure 4: Model-based planning results on LIBERO-LONG, FMB, cloth folding, and cloth unfolding. All of the flows, images, and values shown are generated by FLIP.

## 5.2 LONG-HORIZON VIDEO GENERATION EVALUATION

**Setup.** In this section, we quantitatively evaluate the long-horizon video generation quality of FLIP compared to other video generation models. We choose the same datasets as in Section 5.1 as well as Bridge-V2 (Walke et al., 2023) as the evaluation benchmarks. Here all videos are longer than 200 frames except for Bridge-V2. For Bridge-V2, we train on 10k videos and test on 256 videos with a resolution of $96 \times 128 \times 3$. We choose two baselines: 1) LVDM (He et al., 2022b), a state-of-the-art text-to-video method for video generation; 2) IRASim (Zhu et al., 2024), a conditional video generation method with the end-effector trajectories as the condition. We use SAM2 (Ravi et al., 2024) to label the end-effector trajectory for IRASim. We choose model-based metrics including Latent L2 loss and FVD (Unterthiner et al., 2018) as well as a computation-based metric PSNR (Hore & Ziou, 2010). Latent L2 loss and PSNR measure the L2 distance between the predicted video and the ground-truth video in the latent space and pixel space, and FVD assess video quality by analyzing the similarity of video feature distributions

**Results.** Table 2 shows the results. We can see that our method consistently outperforms baselines in all datasets. LVDM performs badly on LIBERO-LONG and FMB, and better on Bridge-V2. This is because the videos in Bridge-V2 are shorter than the previous two benchmarks. IRASim performs better than LVDM, which shows the importance of trajectory guidance. However, it generates long-horizon videos in an auto-regressive manner, which has worse results than our method, showing that model-based planning can also help generate high-quality videos by concatenating short-horizon videos generated with rich flow guidance. The results on the FMB benchmark are the worst for all methods. This is because the training videos have many discontinuous transitions, where the robot gripper instantly moves to where the next stage begins. Since our model leverages history observations as input conditions, it can sometimes overcome this discontinuous gap. We qualitatively show the model-based planning results on the four tasks in Figure 4.

Since FLIP is a universal framework for all manipulation tasks as long as they have language-annotated video datasets, here we qualitatively show FLIP can be used for complex long-horizon video generation including the ALOHA tasks (Aldaco et al., 2024), pen spinning (Wang et al., 2024), robot pilling (Chen et al., 2024), tying plastic bags (Gao et al., 2023), and human peeling eggs, as shown in Figure 7. More video demos are on our website.

## 5.3 PLAN-GUIDED LOW-LEVEL POLICY

**Setup.** In this evaluation we explore how the generated flow and video plans can be used as conditions for training a manipulation policy to accomplish the task. We aim to answer the question: which one, flow or video (or both at the same time), is more suitable to be used as the condition to guide the learning of the underlying strategy? We use LIBERO-LONG (Liu et al., 2024a) for evaluation, where for each task in LIBERO-LONG, we use 10 demonstrations with action labels and 50 demonstrations without action labels, as done in the baseline method ATM (Wen et al., 2023). During inference, FLIP is a close-loop policy that will replan after every action chunking. We compare FLIP to ATM Wen et al. (2023) and its diffusion-policy version. We also compare OpenVLA Kim et al. (2024) (with both zeros-shot and fine-tuned version) and policies with pretrained FLIP on LIBERO-90 as the planner. Please see Appendix B.2 for these results.

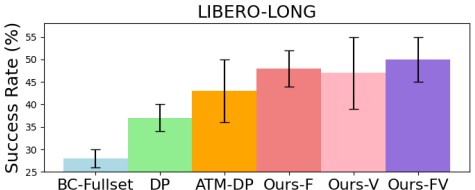

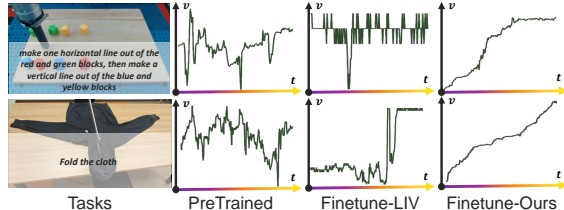

Figure 5: Success rates of different low-level policies on LIBERO-LONG.

Figure 6: Value curves from the pretrained LIV, fine-tuned by LIV, and fine-tuned by FLIP.

|  | LIBERO-10 | | | Language-Table | | | Bridge-V2 | | |
|---|---|---|---|---|---|---|---|---|---|
|  | Latent L2 ↓ | FVD ↓ | PSNR ↑ | Latent L2 ↓ | FVD ↓ | PSNR ↑ | Latent L2 ↓ | FVD ↓ | PSNR ↑ |
| LVDM He et al. (2022b) | 0.366 | 109.41 | 18.852 | 0.364 | 124.75 | 19.943 | 0.328 | 111.34 | 18.104 |
| IRASim (Zhu et al., 2024) | 0.307 | 92.76 | 19.205 | 0.335 | 132.56 | 18.156 | 0.318 | 107.89 | 19.967 |
| FLIP-SC | 0.271 | 89.77 | 20.089 | 0.304 | 137.89 | 18.904 | 0.316 | 127.65 | 18.375 |
| FLIP(Ours) | **0.197** | **27.62** | **28.602** | **0.159** | **21.23** | **33.632** | **0.171** | **38.41** | **34.576** |

Table 3: Quantitative results on short-horizon video generation.

**Results.** The results are in Figure 5. We can see that our plan-guided policies achieve higher success rates than diffusion policies and ATM-DP, showing that dense flow information and high-quality future videos are better to be used as conditions than sparse flow information. The flow-video-guided policy (Ours-FV) achieves the best average success rates across all methods, showing the advantage of using multi-modality information as conditions. Although video-guided policies (Ours-V) achieve a competitive mean success rate, they have a high variance, showing that using videos as conditions is unstable. This may come from that the generated future videos can become low-quality if the robot deviates from the trained trajectories. Instead, with the flow as extra conditions, the variance becomes lower, showing the stability of dense image flow predictions.

## 5.4 EXPERIMENTS ON FUNDAMENTAL MODULES OF FLIP

**Action Module Experiments.** We use two metrics to assess the flow generation model $\pi_f$ quantitatively (Jiang et al., 2024b): 1) Average Distance Error (ADE) between the generated and the ground truth flows in pixel units on all query points; 2) Less Than Delta Ratio (LTDR): the average percentage of points within the dis-

|  | LIBERO-10 | | Bridge-V2 | |
|---|---|---|---|---|
|  | ADE ↓ | LTDR ↑ | ADE ↓ | LTDR ↑ |
| ATM (Wen et al., 2023) | 19.6 | 53.8% | 18.4 | 66.1% |
| Ours-ABS | 20.5 | 57.3% | 17.9 | 59.3% |
| Ours-NoAUX | 14.5 | 73.2% | 12.7 | 75.6% |
| Ours | **12.7** | **76.5%** | **11.9** | **80.2%** |

Table 4: Quantitative results of the action model.

tance threshold of 1, 2, 4, and 8 pixels between the reconstructed and the ground truth flows at each time step. Since most of the points are stationary points, in order to better demonstrate the results, we only calculate points with $\delta_s \geq 1$. We also do experiments that compare using CVAE and diffusion models as the action module in Appendix B.3.

We use LIBERO-LONG (Liu et al., 2024a) and Bridge-V2 (Walke et al., 2023) for evaluation. We compare our method with 3 baselines: 1) ATM (Wen et al., 2023), the state-of-the-art flow prediction module for manipulation tasks; 2) Ours-ABS: directly generating absolute flow coordinates at each timestep rather than generating the scale and direction; 3) Ours-NoAUX: the same architecture of ours with no auxiliary training losses (the flow and image reconstruction losses).

From Table 4, we can see that Ours-ABS generally achieves the same results as ATM, and predicting the scale and directions are better than ATM and Ours-ABS, showing that directly regressing the absolute coordinates is worse than predicting the delta of flows at each timestep. We can also see that the auxiliary losses can help improve the final results.

**Dynamics Module Experiments.** We evaluate our dynamics module separately with the ground truth flows as conditions on *short-horizon* video generation. We use PSNR (Hore & Ziou, 2010), latent L2 loss, and FVD (Unterthiner et al., 2018) as metrics. We use LIBERO-LONG (Liu et al., 2024a), Bridge-V2 (Walke et al., 2023), and Language-Table (Lynch et al., 2023) as the evaluation datasets. We use three baselines (as introduced in Section 5.2): 1) LVDM (He et al., 2022b); 2) IRASim (Zhu et al., 2024); 3) Ours-SC: using AdaLN-Zero for all kinds of conditions.

Results are in Table 3. The result trends across methods are generally consistent with the long-horizon video generation results in Table 2. FLIP-SC generally achieves the same performance with

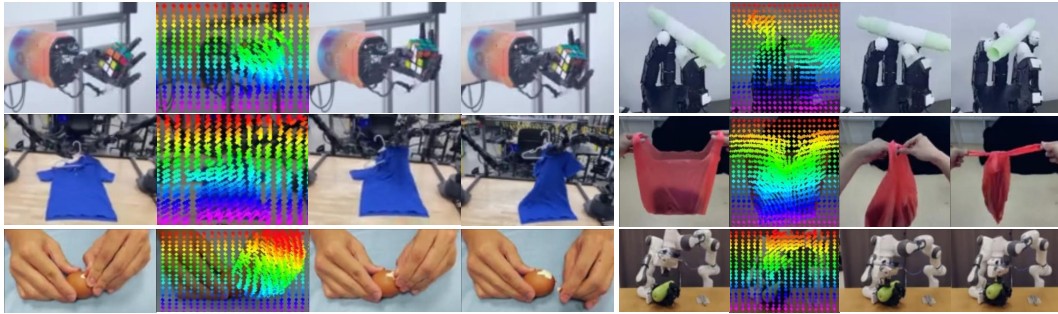

Figure 7: FLIP is a general framework for diverse kinds of manipulation tasks across objects and robots, even for human hands. All of the flows and images are generated.

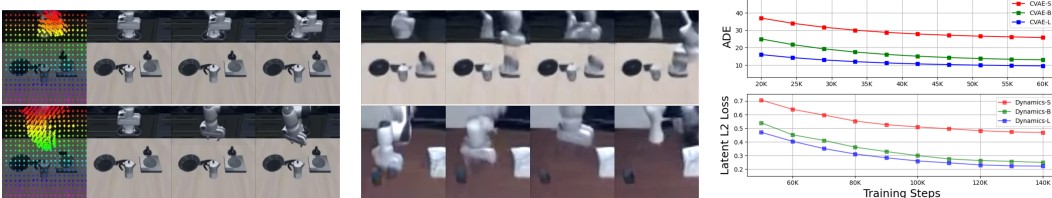

Figure 8: Interactive ability.     Figure 9: Zero-shot transfer.     Figure 10: Scalability.

IRASim, showing that even if the model is given dense flow information, it requires a fine-grained mechanism to leverage the condition for video generation.

**Value Module Experiments.**   We here qualitatively show the fine-tuned value curves of our method compared to the original LIV (Ma et al., 2023) method on two different tasks consisting of Language-Table (Lynch et al., 2023) and cloth folding in Figure 6. We also show the value curves before fine-tuning. We can see our method consistently gets smoother value curves than the original LIV method, where the value curves have violent oscillations.

## 5.5 APPLICATIONS AND SCALING

Finally, we train FLIP on LIBERO-90, a large-scale simulation manipulation dataset to show three properties of FLIP. We use 50 videos for each task in the resolution of $3 \times 64 \times 64$.

**Interactive World Model.**   We first show that the trained dynamics module is interactive: it can generate corresponding videos given image flows specified by humans. We use SAM2 (Ravi et al., 2024) to select the region of the robot arm and manually give flows in different directions. Results are shown in Figure 8. We can see the robot arm can move left or right according to the given flow.

**Zero-Shot Generation.**   Secondly, we show that the trained FLIP has zero-shot transfer ability. We test the trained model on LIBERO-LONG. Results are shown in Figure 9. Interestingly, we can see that the pretrained model, without fine-tuning, can generate natural movement for the robot arm with unseen observations and instructions. This shows FLIP has a certain knowledge transfer ability.

**Model Scaling.**   We show that the action and dynamics module are scalable with increasing model sizes. Figure 10 shows the smoothed ADE and Latent L2 loss on the validation set. It shows that increasing the model size can consistently help achieve better performance for both modules.

## 6 CONCLUSION AND LIMITATION

In this work, we present FLIP, a flow-centric generative planning method for general-purpose manipulation tasks. FLIP is trained on only video and language data, can perform model-based planning on the trained world model to synthesize long-horizon plans, and can guide low-level policy learning. FLIP has the potential to scale up with increasing data and computation budgets in the future.

A major limitation of FLIP is the slow speed of planning, which is restricted by extensive video generation processes during the planning phase. This restricts our method on quasi-static manipulation tasks. Another limitation is that FLIP does not use physical properties and 3D information of the scene. Future works can develop physical 3D world models and extend FLIP to 3D scenarios.

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

# A METHOD DETAILS

## A.1 LATENT DIFFUSION MODELS

**Diffusion Models.** Diffusion models (Song et al., 2020; Ho et al., 2020) typically contain a forward nosing process and a reverse denoising process. During the forward process, we gradually apply noise to real data $x_0$: $q(x_t|x_0) = \mathcal{N}(x_t; \sqrt{\bar{a}_t}x_0, (1 - \bar{a}_t)\mathbf{I})$ over $T$ timesteps, where constants $\bar{a}_t$ are hyperparameters. By applying the reparameterization trick, we can sample $x_t = \sqrt{\bar{a}_t}x_0 + \sqrt{1 - \bar{a}_t}\epsilon_t$, where $\epsilon_t \sim \mathcal{N}(0, \mathbf{I})$. During the reverse process, it starts from Gaussian noise $x_T \sim \mathcal{N}(0, \mathbf{I})$ and gradually removes noises to recover $x_0$: $p_\theta(x_{t-1}|x_t) = \mathcal{N}(\mu_\theta(x_t), \Sigma_\theta(x_t))$. With reparameterizing $\mu_\theta$ as a noise prediction network $\epsilon_\theta$, the model can be trained with $\mathcal{L}_{simple}(\theta) = ||\epsilon_\theta(x_t) - \epsilon_t||_2^2$. We also learn the covariance $\Sigma_\theta$ following Nichol & Dhariwal (2021); Peebles & Xie (2023) with the full KL loss.

**Latent Diffusion and Tokenization.** Latent diffusion models (Rombach et al., 2022; Ma et al., 2024) perform diffusion process in a low-dimensional latent space $z^{ld}$ rather than the original pixel space. We leverage the pre-trained VAE in SDXL (Podell et al., 2023) to compress each frame $o_t$ to latent representations: $z_t^{ld} = Enc(o_t)$, and after the denoising process, the latent representation can be decoded back to the pixel space with the VAE decoder: $o_t = Dec(z_t^{ld})$. For each $z^{ld}$, it is divided into image patches and tokenized by convolutional networks to $P$ tokens with $D$ dimensions (hidden size). Sequencing the image tokens of all $T$ frames, we get the video token in the shape of $T \times P \times D$.

**Spatial-Temporal Attention Mechanism.** We leverage transformers (Vaswani, 2017) to implement the dynamics module, and use the memory-efficient spatial-temporal attention mechanism (Ma et al., 2024; Bruce et al., 2024; Zhu et al., 2024), where each attention block consists of a spatial attention block and a temporal attention block. The spatial attention operates on the $1 \times P$ tokens within each frame, and the temporal attention operates on the $T \times 1$ tokens across $T$ timesteps at the same location.

## A.2 LANGUAGE-VISION REPRESENTATION

The original fine-tuning loss $\mathcal{L}_{LIV} = \mathcal{L}_I(\psi_I) + \mathcal{L}_L(\psi_I, \psi_L)$ is calculated on sampled sub-trajectory batch data $\left\{o_s^i, \ldots, o_k^i, o_{k+1}^i, \ldots, o_T^i, g^i\right\}_{i=1}^B$ from each task $\mathcal{T}_i$, where $s \in [0, T_i - 1]$, $s \leq k < T_i$. They have the following forms:

$$\mathcal{L}_I(\psi_I) = \frac{1-\gamma}{B}\sum_{i=1}^B[-\mathcal{S}(\psi_I(o_s^i), \psi_I(o_T^i))] + \log\frac{1}{B}\sum_{i=1}^B \exp[\mathcal{S}(\psi_I(o_k^i), \psi_I(o_T^i)) + 1 - \gamma\mathcal{S}(\psi_I(o_{k+1}^i), \psi_I(o_T^i))],$$

$$\mathcal{L}_L(\psi_I, \psi_L) = \frac{1-\gamma}{B}\sum_{i=1}^B\left[-\log\frac{e^{(1-\gamma)\mathcal{S}(\psi_I(o_T^i), \psi_L((g^i))}}{\frac{1}{B}\sum_{j=1}^B\left[e^{(1-\gamma)\mathcal{S}(\psi_I(o_T^j), \psi_L(g^i))}\right]}\right],$$

$$(2)$$

## A.3 LOW-LEVEL POLICY

In this work, we use a low-level policy with a similar structure to diffusion policy (Chi et al., 2023). We show the architecture of this policy in Figure 11. We employ a spatial-temporal attention mechanism. Specifically, the input contains the agent view observation history $o_{t-12:t}^a \in \mathbb{R}^{12 \times 3 \times 128 \times 128}$ and the eye in hand observation history $o_{t-12:t}^e \in \mathbb{R}^{12 \times 3 \times 128 \times 128}$ at timestep $t$, the proprioception state history $s_{t-12:t} \in \mathbb{R}^{12 \times 123}$, the language tokens extracted from Meta Llama 3.1 8B (Dubey et al., 2024) $g \in \mathbb{R}^{T_g \times 4096}$, the predicted flow for both the agent view $\mathbf{p}_{t:t+16}^a \in \mathbb{R}^{16 \times 529 \times 2}$ and eye in hand view $\mathbf{p}_{t:t+16}^e \in \mathbb{R}^{16 \times 529 \times 2}$, and the predicted future videos for both the agent view $\hat{o}_{t:t+16}^a \in \mathbb{R}^{16 \times 3 \times 128 \times 128}$ and eye in hand view $\hat{o}_{t:t+16}^e \in \mathbb{R}^{16 \times 3 \times 128 \times 128}$. The low-level policies are a denoising model that will predict the gradient field of the action chunking $\nabla E(A_t)$, and after 500 denoising steps, we can get the future action sequences $a \in \mathbb{R}^{16 \times 7}$ where 7 is the action size. We use the action in a receding horizon manner where we only execute 8 steps and we replan the future flow and videos and predict another 16 action steps iteratively.

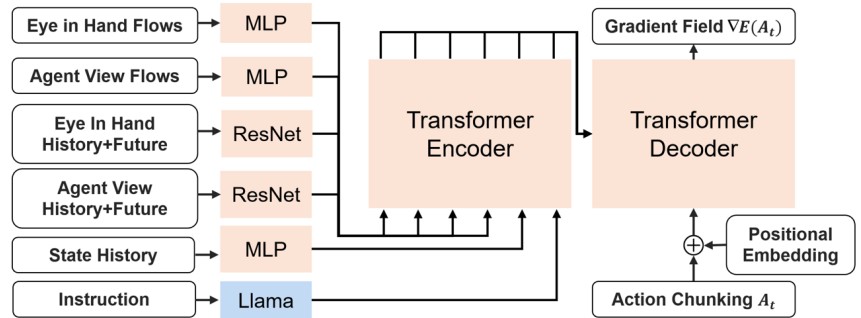

Figure 11: Low-level policy architecture. This is Ours-FV. Ours-F and Ours-V are in the same architecture without corresponding input conditions.

## B  EXPERIMENT DETAILS

### B.1  TRAINING DETAILS

We report the hyperparameters of the models we trained in Table 5 and Table 6. We train all data with observation history equals to 16 and future flow horizon equals to 16.

|  | CVAE-S | CVAE-B | CVAE-L |
|---|---|---|---|
| Encoder Layer | 4 | 6 | 8 |
| Decoder Layer | 6 | 8 | 12 |
| Hideen Size | 384 | 768 | 1024 |
| Learning Rate | 1e-4 | 5e-5 | 1e-5 |
| Image Patch Size | 8 | 8 | 8 |
| Head Number | 4 | 8 | 12 |

Table 5: Hyperparameters of CVAE.

|  | D-S | D-B | D-L |
|---|---|---|---|
| Layers | 8 | 12 | 16 |
| Hideen Size | 384 | 768 | 1024 |
| Learning Rate | 1e-4 | 1e-4 | 1e-4 |
| Head Number | 6 | 12 | 16 |

Table 6: Hyperparameters of the dynamics module.

### B.2  MORE POLICIES RESULTS

Besides the low-level experiments in Section 5.3, we also perform experiments in the LIBERO-LONG task suite with two other models:

1. OpenVLA Kim et al. (2024): this is a large-scale retained that is designed for general-purpose vision-language-action prediction. We use the pretrained checkpoint from the official paper and test with both zero-shot and fine-tuned models. We use 50 demonstrations for each task in LIBERO-LONG to fine-tune the OpenVLA model. For consistency, we use a resolution of 128×128 for fine-tuning. For fair comparisons, we only use the agent view images as input because the original OpenVLA is only trained with third-view images.

2. Pretrained FLIP on LIBERO-90 and fine-tuned on LIBERO-10 as the high-level planner. In this setting, we train FLIP on 50×90 action-less demonstrations with a resolution of 64×64 from LIBERO-90, and finetune it with 50×10 from LIBERO-LONG, and use this FLIP model as the planner to train the same low-level policy as in the main paper. Here we only test the flow-conditioned policy version, and call it Ours-90.

**Results.** We show the results together with the main paper results in Figure 12. We can see that OpenVLA cannot handle the long-horizon tasks of LIBERO-LONG either with zero-shot or fine-tuned models, showing there is still a long way to go for general-purpose vision-language-action models. Ours-90 performs similarly to Ours-F and Ours-FV, showing that pretraining in other tasks may not bring significant improvement for low-level policy learning. This comforts with the life-long learning results in the original LIBERO paper (Liu et al., 2024a), where they also show that pretraining cannot help (sometimes even hurt) the policy training results.

It is worth noting that, in the original OpenVLA paper, they also fine-tuned the pretrained model on LIBERO-LONG tasks and archived a 53.7 ± 1.3% success rate. We think the success of their results comes from two aspects, which cannot be true in our setting: 1) we are using a resolution of 128×128, which may not be large enough to represent the details in the scene. In comparison, OpenVLA uses a resolution of 256×256. 2) We are using the official demonstrations provided by the LIBERO paper, which may not be as good as the re-collected demonstrations in their demonstrations.

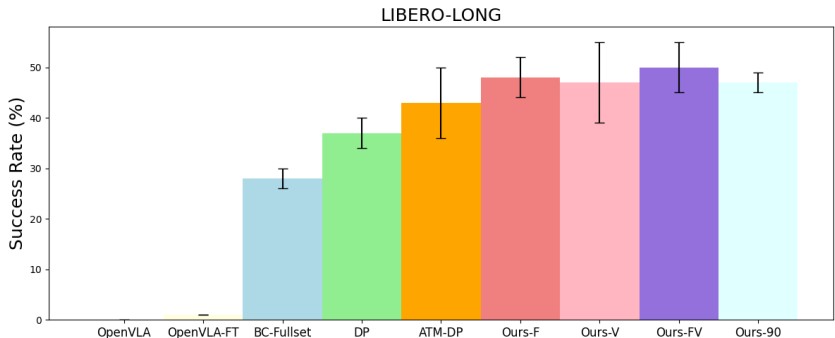

Figure 12: Low-level policy results in LIBERO-LONG.

### B.3 ABLATION STUDIES OF ACTION MODULE

To verify if diffusion models can generate better results for image flow prediction (the action module of FLIP) than a CVAE model, here we design a diffusion model for the flow generation task. We use the same architecture as the CVAE decoder for the diffusion model (as illustrated in Figure 2, where a transformer encoder takes as input the query points tokens, the image history tokens, the language embedding tokens, and noisy inputs, and outputs the scale and direction gradient fields for each query point. The network structure is shown in Figure 13. We perform comparison experiments in both LIBERO-LONG and Bridge-V2 data.

**Results.** Table 7 shows the results. We can see that CVAE performs better on LIBERO-LONG tasks and the diffusion model performs better on Bridge-V2 tasks, but their performance difference is minor. We can see the main improvement of the action module comes from: 1) predicting the delta action (scale and direction) rather than predicting the absolute coordinates of the image flows; 2) auxiliary losses as stated in our main paper.

Although using the diffusion model may perform better in some tasks, the inference speed is slower than CVAE models. Given that their performance is not too different, we still use CVAE as our default action module.

### B.4 REAL WORLD EXPERIMENTS

In order to show the application of FLIP on real-world tasks, here we design two long-horizon real-world manipulation tasks for testing. We use a 6-DOF X-arm as the robot arm, and use two RealSense D435i cameras to get the visual inputs. We put one camera on the other robot arm as the third-view camera and another camera on the wrist of the robot arm as the eye-in-hand camera. The control frequency of both tasks is set to 10Hz. We also train a diffusion policy and an ATM (Wen

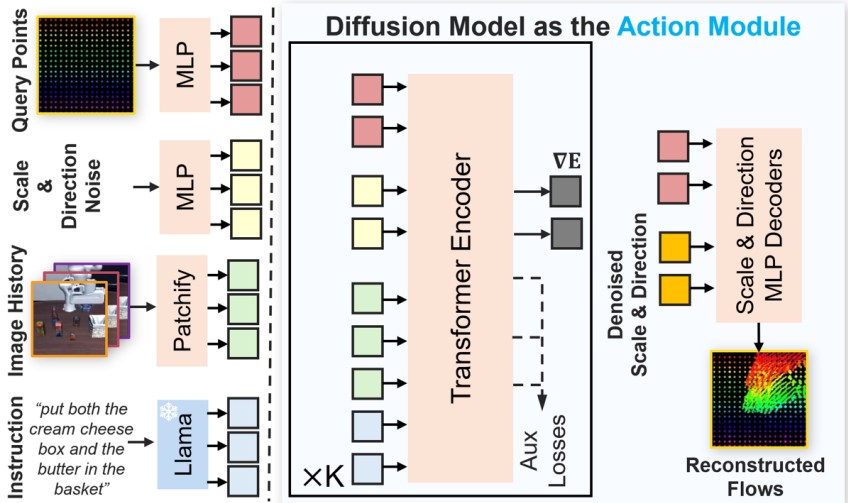

Figure 13: Using diffusion model for the action module. The model also generates the scale and directions for each query point and the full flow trajectory can be reconstructed by them.

| | LIBERO-LONG | | Bridge-V2 | |
|---|---|---|---|---|
| | ADE ↓ | LTDR ↑ | ADE ↓ | LTDR ↑ |
| ATM (Wen et al., 2023) | 19.6 | 53.8% | 18.4 | 66.1% |
| Ours-ABS | 20.5 | 57.3% | 17.9 | 59.3% |
| Ours-NoAUX | 14.5 | 73.2% | 12.7 | 75.6% |
| Ours(CVAE) | **12.7** | **76.5%** | 11.9 | 80.2% |
| Diffusion | 13.4 | 75.9% | **10.2** | **82.7%** |

Table 7: Comparison results of CVAE and Diffusion models as the action module for FLIP.

et al., 2023) policy as the baseline for each task. We perform experiments in a table-top setting, as shown in Figure 14. We test both tasks with 20 random initialized episodes. The two tasks are:

1. Tea Scooping Task. In this task, there is a white bowl of dry tea leaf (Tieguanyin), a yellow bowl with an iron spoon, and an empty cup on the table, as shown in Figure 15. The robot is trained to first pick up the spoon from the yellow bowl, then use the spoon to scoop the tea

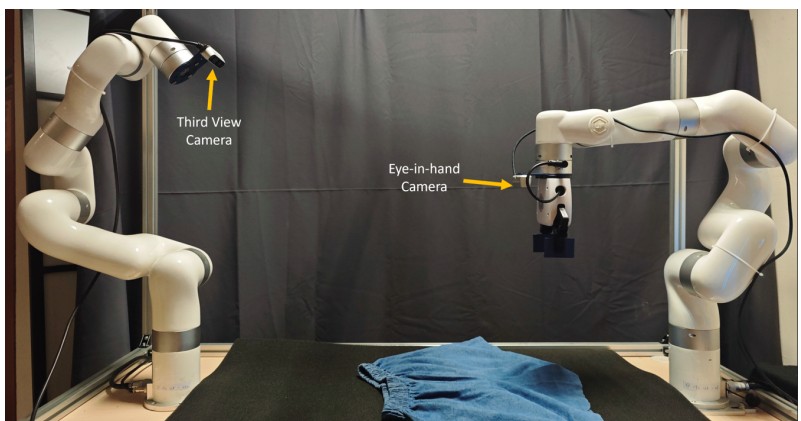

Figure 14: Our real-world experiment setting. We use the right X-arm as the manipulator, and use two RealSense D435i cameras as visual inputs.

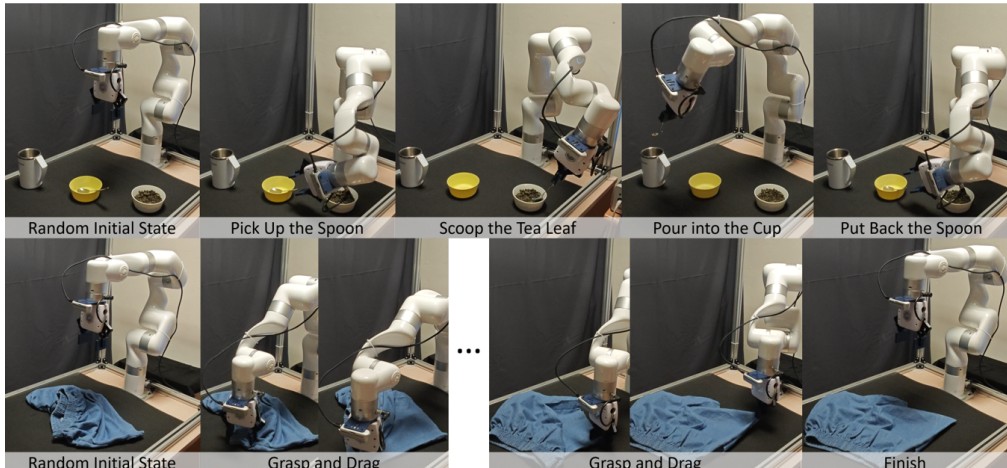

Figure 15: Two tasks in our real-world experiments.

leaf from the white bowl pour the tea leaf into the cup, and finally put back the spoon into the yellow bowl. The positions of both the bowls and the cup are randomized in the space that the robot arm can reach. The task is successful if all three stages are accomplished. The action space of the robot is 7-dim, including the 6-DOF pose and the gripper action. We collect 40 demonstrations with a scripted policy to train FLIP and a flow-guided low-level policy. We choose this task because it is both long-horizon and requires precise grasping and manipulation. It also requires tool use and interaction between rigid body and granular objects.

2. Cloth Unfolding Task. In this task, a pair of denim shorts was randomly thrown on the table, and the robot is trained to unfold it to make it flat, as shown in Figure 15. The task is successful if the projection area of the jeans on the table is greater than 85% of its maximum area. The action space of the robot is 6-dim, which consists of a 2-dim end-effector coordinate in the x-y plane, a 1-dim grasping orientation angle, a 2-dim dragging coordinate in the x-y plane, and the gripper aperture. We fix the grasping height to 22 mm for this task. We collect 50 demonstrations for this task with a scripted policy. We choose this task because it is both long-horizon and difficult because of the deformation of the jeans, which requires a long-horizon task planning ability for the policy. The robot has to finish the task in 15 grasps and drags.

**Results.** Table 8 shows the results of both tasks. We can see that our policy is significantly better than the baselines. These two tasks are very difficult in out setting since we only use no more than 50 demonstrations for each of them, while modern image-based imitation learning algorithms (such as Diffusion Policy (Chi et al., 2023)) usually require more than 500 demonstrations for such kinds of long-horizon tasks. In contrast, our method benefits from the high-level model-based planning advantages and can correct back to the good trajectory when it deviates from it, thus the success rates are improved.

| | Diffusion Policy (Chi et al., 2023) | ATM (Wen et al., 2023) | Ours |
|---|---|---|---|
| Tea Scooping | 15.0% | 25% | **55%** |
| Cloth Unfolding | 0.0% | 0.0% | **50.0%** |

Table 8: Success rates of real robot experiments over 20 evaluations.

## B.5 DATA SCALABILITY

To show the performance change of FLIP along with different amounts of training data, here we perform an experiment on LIBERO-LONG to show the planning success rates change with different

amount of demonstrations. Results are shown in Table 9. From the results, we can see that with more data for each task, the planning success rates become better.

| | LIBERO-LONG (Liu et al., 2024a) |
|---|---|
| FLIP-10 | 0% |
| FLIP-20 | 5% |
| FLIP-30 | 40% |
| FLIP-40 | 90% |
| FLIP-50 | **100%** |

Table 9: Planning results change along with data amounts for LIBERO-LONG. The number shown in the left column is the demonstration number for each task during training.

### B.6 GENERATIVE PLANNING WITH VISUAL DISTRACTION

In this section, we perform a qualitative experiment to see how will our FLIP be affected in the presence of noise or visual obstructions. To this end, we manually add an image of an apple in the initial image of the LIBERO-LONG tasks, as shown in Figure 16 and Figure 17, and see how FLIP performs generative planning in this setting.

**Results.** We add the visual distraction image in different areas of the initial image, and we can see that, the video generation model will first fail after several planning steps and generative distorted cups and make the whole image grey. However, before the model fails, the flow generation model still performs very well, which shows that the flow generation model can resist visual distractions, while the video generation model is susceptible to visual distractions.

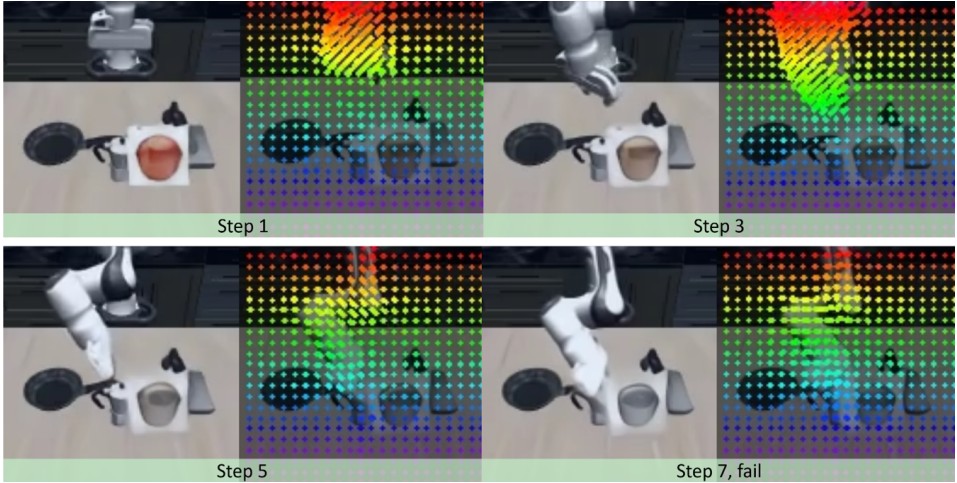

Figure 16: The apple is put on the target cup, and the generative planning fails in 7 steps (the cup is distorted, by which we call the planning fails). Note, the flow model performs well before the image is distorted and turns to grey, which shows that the flow model can capture the geometry of the objects and understand the physical movement requirements of the task.

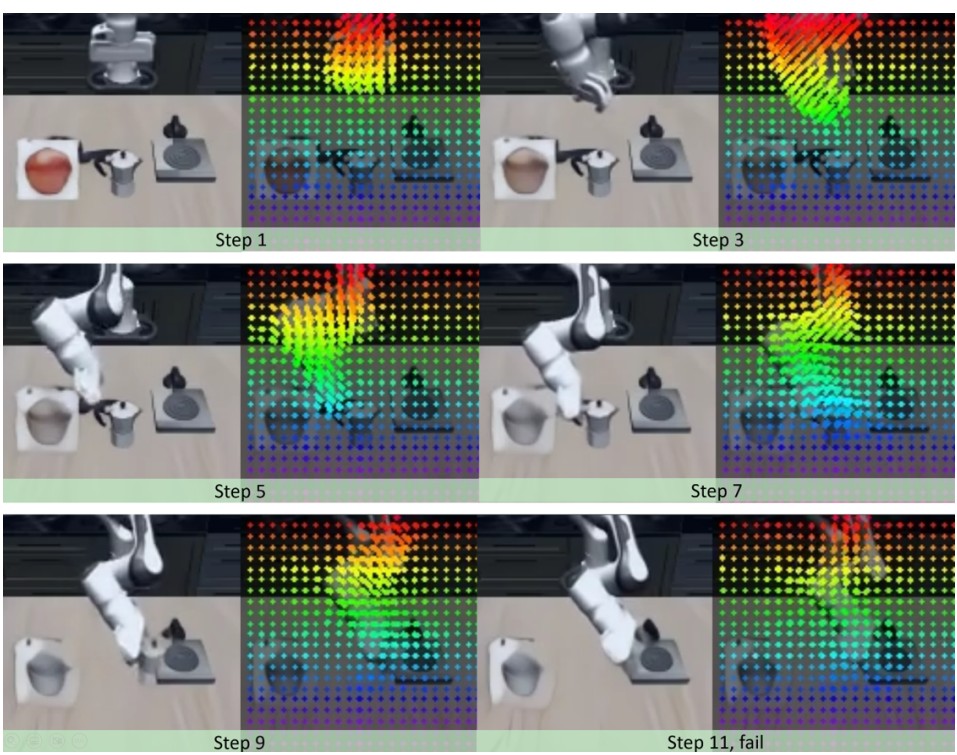

Figure 17: The apple is put away from the target object, and the generative planning can exist more steps.

