# OpenReview forum: "FLIP: Flow-Centric Generative Planning as General-Purpose Manipulation World Model"
_ICLR.cc/2025/Conference — ICLR 2025 Poster_

### Official Review · Reviewer_6B9z · 2024-10-30

**Soundness:** 3
**Presentation:** 3
**Contribution:** 3
**Rating:** 6
**Confidence:** 3

**Summary:**

The paper introduces a model-based planning framework (FLIP) designed for general-purpose robotic manipulation tasks using language and vision inputs. The framework allows for the progressive synthesis of long-horizon action plans, starting from an initial image and language instruction. FLIP effectively uses image flows to represent complex movements, enhancing the planning process for manipulation tasks across various objects and robots. They propose a multi-modal flow generation model predicting actions by generating dynamic representations of movements, simulating future video sequences based on the proposed actions, and evaluate the generated videos to maximize the task's success probability.

**Strengths:**

- The paper is well-written and methodically presented.
- FLIP introduces a novel approach to model-based planning by integrating multi-modal inputs, which enhances its versatility for various manipulation tasks
- FLIP is designed to scale with increasing model and data budgets, making it suitable for a variety of applications and capable of leveraging more extensive datasets as they become available.
- The generated plans can be used to inform and train low-level control policies which can support several hierarchical policies that require strategic decision-making and planning.

**Weaknesses:**

- The framework may struggle with ambiguities in language instructions or unexpected changes in the environment.
- Depending on the computational demands of the multi-modal modules, real-time performance in dynamic environments could be a concern. Evaluating the speed and efficiency of planning under real-time constraints would be crucial.

**Questions:**

- How does FLIP compare to other state-of-the-art planning frameworks in terms of efficiency and effectiveness?
- What are the limitations of using image flow as an action representation? In which scenarios/tasks, this would not be ideal?
- How does FLIP handle ambiguities in language instructions?
- Are there any results on the real robot that show the applicability on world models even for quasi-static tasks?
- How will the model's performance be affected in the presence of noise or visual obstructions?
- How does the performance of FLIP depend upon or scale with the size of the available data, numerical analysis for the same would be helpful to understand the efficiency of the proposed approach.

Please also address other comments in the weaknesses section above.

---

> ### Author Response · Authors · 2024-11-26
> **Response to Reviewer 6B9z**
>
> We thank reviewer 6B9z again for reviewing our paper and providing helpful feedback! Here we
> address your concerns one by one.
>
> > The framework may struggle with ambiguities in language instructions or unexpected changes in the environment.
>
> We point out that ambiguous language instruction and unexpected environmental changes fall outside the scope of this study. This work is based on the static MDP assumption, where we assume a language instruction can clearly specify the goal, and the MDP is not changed during the model-based planning and policy execution.
>
> > Depending on the computational demands of the multi-modal modules, real-time performance in dynamic environments could be a concern. Evaluating the speed and efficiency of planning under real-time constraints would be crucial.
>
> Thanks for your question. Yes, real-time (which means about 30 HZ) inference speed is a question of our model. This is mainly constrained by: 1) the denoising process of the dynamics module; 2) the model-based planning process with three networks. However, with the development of faster denoising techniques such as [1], we believe we can achieve near real-time planning performance in the future.
>
> [1] Lu, Cheng, and Yang Song. "Simplifying, Stabilizing and Scaling Continuous-Time Consistency Models." arXiv preprint arXiv:2410.11081 (2024).
>
> > How does FLIP compare to other state-of-the-art planning frameworks in terms of efficiency and effectiveness?
>
> We compare FLIP to two generative planning methods (UniPi [1] and IRASim [2]) in Table 1 and Table 2. We hope these results can address your concerns.
>
> > What are the limitations of using image flow as an action representation? In which scenarios/tasks, this would not be ideal?
>
> Thanks for your good question! There are two main limitations of using image flows as action representation: 1) 3D motions that have very small movements in the 2D space. For example, an arrow flying in the direction perpendicular to the screen. 2) objects with a very small size in the image. For example, if some object is put very far away from the camera, there could be no query points sampled on that object because it only occupies several pixels. Thus its movement will not be predicted.
>
> > How does FLIP handle ambiguities in language instructions?
>
> Currently, we use the same language instruction during testing.
>
> The language embedding is extracted from Meta Llama 3.1, thus we leave the ambiguities in language instructions to LLMs. We show zero-shot results in our website of the pretrained FLIP model on LIBERO-90 to LIBERO-LONG tasks, where the language instruction and the whole scene are new to the model. Since all of the three modules of FLIP is scalable, we hope in the future, with internet-level data, it can handle the ambiguities in language instructions.
>
>
> > Are there any results on the real robot that show the applicability on world models even for quasi-static tasks?
>
> Thanks for your question. We add real robot experiments in Appendix B.4.
>
> > How will the model's performance be affected in the presence of noise or visual obstructions?
>
> Thanks for your good question! We add visual obstructions experiments in the LIBERO-LONG experiments. We manually add a picture of an apple into the scene during the planning, and from the results, we can see that our flow generation model can resist visual distraction, and the video generation model can generate dynamics for a few planning steps (x16 for execution steps), while it will generation distorted images after that. This shows that flows are a good choice to represent high-level actions that can be robust to visual distractions. Since our model is only trained on specific sense data, we believe with internet-level data, the video generation model will also perform better.
>
> > How does the performance of FLIP depend upon or scale with the size of the available data, numerical analysis for the same would be helpful to understand the efficiency of the proposed approach.
>
> Thanks for your question. We add a data scalability experiment in Appendix B.5. From Table 9, we
> can see that with more data for each task, the planning success rates become better.

---

> ### Author Response · Authors · 2024-11-26
>
> We thank reviewer 6B9z again for reviewing our paper and providing helpful feedback! We hope the additional experiments and clarifications have addressed the reviewer’s concerns. If the reviewer has any further concerns or questions, please let us know before the rebuttal period ends, and we will be happy to address them.

---

> ### Comment · Reviewer_6B9z · 2024-11-27
>
> The authors response clarify my concerns and I update my score to 6.

---

> > ### Author Response · Authors · 2024-11-27
> >
> > Thanks for your reply! We deeply appreciate your thorough review and constructive comments, which have been invaluable in refining the paper's presentation.
> >
> > We have carefully revised the manuscript to address the concerns that led to your lowered score, and we sincerely hope the changes will allow for a more favorable consideration. Thanks!

---

### Official Review · Reviewer_R3ys · 2024-11-02

**Soundness:** 3
**Presentation:** 3
**Contribution:** 3
**Rating:** 6
**Confidence:** 4

**Summary:**

This paper proposes FLIP, a model-based planning algorithm on visual space.
The algorithm consists of three main modules 1) a flow generation model as an action proposal module, 2) a flow-conditioned video generation model as a dynamic module, and 3) a value module.
The flow generation model generates flows based on the observation and a language instruction.
The video generation model generates videos with a Diffusion Transformer based the predicted flows.
The value module, built on a vision-language representation model, is used to assign values for each frame in a video to enable model-based planning in the image space.
Model-based planning algorithm leverages the three modules to search for a sequence of flow actions and video plans that maximizes the discounted return.
Experiments were performed to evaluate the capability of the proposed method on generating video plans, generating long-horizon videos, and accomplishing manipulation tasks.
The proposed method outperforms comparing baseline methods and showcases interactive properties, zero-shot transfer capability, and scalability.

**Strengths:**

The paper is well-written and clear.
Leveraging CVAE to address the multi-modality of flows is well-motivated.
Additionally, the paper proposes to use a mixed-conditioning mechanism for multi-modal conditional video generation.
For the value module, the paper modifies the original LIV method and uses video clips instead of video frames as a unit to account for the noisy value prediction.
Experiments show that the proposed method surpasses comparing baseline methods in generating video plans, long-horizon videos, and performing manipulation tasks.
The paper also showcases interesting interactive properties, zero-shot transfer ability, and scalability of the proposed method.

**Weaknesses:**

1. There are no real-robot experiments to validate the effectiveness of the proposed method in policy learning in the real world. Incorporating such experiments and comparing with recent policy learning methods (e.g. ATM, OpenVLA [1] or Octo [2]) would provide a more comprehensive understanding of FLIP's performance in real-world policy learning.

2. The paper only evaluates the policy learning capability on a suite of LIBERO, i.e. LIBERO-Long. Conducting an evaluation to assess the generalization capability of the proposed method, e.g. on the LIBERO-Object suite, would be beneficial. Also, including a comparison with recent policy learning methods, like OpenVLA [1] or Octo [2], would further strengthen the paper.

2. In Sec. 5.2, the paper compares with IRASim on a text-to-video task. However, IRASim generates video based on a trajectory instead of a text. For the text-to-video task, it would be better to compare with a text-to-video method (e.g. [3]).

[1] Kim, Moo Jin, et al. "OpenVLA: An Open-Source Vision-Language-Action Model." arXiv preprint arXiv:2406.09246 (2024).

[2] Team, Octo Model, et al. "Octo: An open-source generalist robot policy." arXiv preprint arXiv:2405.12213 (2024).

[3] Ma, Xin, et al. "Latte: Latent diffusion transformer for video generation." arXiv preprint arXiv:2401.03048 (2024).

**Questions:**

1. In Sec. 5.1, the paper compares with FLIP-NC, an ablation of the proposed method FLIP which has no value module as guidance. Is it possible to provide more details on how FLIP-NC performs beam search without a value guidance?

2. Is it possible to provide a more detailed description on the typical failure modes of FLIP in policy learning (Sec. 5.3) ?

3. In the Dynamics Module Experiments in Sec. 5.4, the paper compares with LVDM and IRASim on short-horizon video generation. The proposed method is provided with ground-truth flows to generate videos. What information is provided for the two comparing baseline methods?

4. Are the flow generation model and the video generation model trained individually or jointly?

---

> ### Author Response · Authors · 2024-11-26
> **Response to Reviewer R3ys**
>
> We thank reviewer R3ys again for reviewing our paper and providing helpful feedback! Here we
> address your concerns one by one.
>
> > There are no real-robot experiments to validate the effectiveness of the proposed method in policy learning in the real world.
>
> We have added real-world experiments in Appendix B.4. Please check this section and our website for more real-world results.
>
> > Including a comparison with recent policy learning methods, like OpenVLA [1] or Octo [2], would further strengthen the paper.
>
> Thanks for your suggestion! We add an experiment of OpenVLA on LIBERO-LONG to compare
> its results with FLIP. We show the results of zero-shot and fine-tuned with 50 demonstrations for
> each task of OpenVLA in Appendix B.2. We can see that OpenVLA cannot handle the long-horizon
> tasks of LIBERO-LONG either with zero-shot or fine-tuned models, showing there is still a long
> way to go for general-purpose vision-language-action models.
>
> It is worth noting that, in the original OpenVLA paper, they also fine-tuned the pretrained model
> on LIBERO-LONG tasks and archived a 53.7 ± 1.3% success rate. We think the success of their
> results comes from two aspects, which cannot be true in our setting: 1) we are using a resolution
> of 128×128, which may not be large enough to represent the details in the scene. In comparison,
> OpenVLA uses a resolution of 256×256. 2) We are using the official demonstrations provided by
> the LIBERO paper, which may not be as good as the re-collected demonstrations in their demonstrations.
>
> > The paper compares with IRASim on a text-to-video task. However, IRASim generates video based on a trajectory instead of a text. For the text-to-video task, it would be better to compare with a text-to-video method (e.g. [3]).
>
> Thanks for your question. The LVDM [4] baseline we used is a text-to-video method. We hope this can answer your question.
>
> > Is it possible to provide more details on how FLIP-NC performs beam search without a value guidance?
>
> Thanks for your question. FLIP-NC performs a beam search in an autoregressive manner, which means it initializes the same number of beams as FLIP and generates the long-horizon flows and videos iteratively within each beam, without multiple action generation for each beam.
>
> > Is it possible to provide a more detailed description on the typical failure modes of FLIP in policy learning (Sec. 5.3) ?
>
> Thanks for your question. We provide some failure videos of the trained policy in our attached videos and on our website. The typical failure mode is that the robot has a small action error when it is going to grasp something, which leads to an unsuccessful grasp and the robot will do this action repeatedly. Interestingly, our flow generation model can still generate reasonable future flows in these out-of-distribution areas to guide the robot to the correct regions. However, it sometimes cannot accomplish this in given episode timestep limits.
>
> > What information is provided for the two comparing baseline methods (LVDM and IRASim)?
>
> Thanks for your question. For LVDM, there is no additional information provided. For IRASim, it is provided with the end-effector trajectory extracted with SAM2.
>
> > Are the flow generation model and the video generation model trained individually or jointly?
>
> They are trained individually.

---

> > ### Comment · Reviewer_R3ys · 2024-11-27
> > **Response to Authors**
> >
> > Thank you for your detailed response. Some of my concerns have been addressed. The updated real-robot experiments are able to showcase the performance of the proposed method in real-world policy learning. Based on the response, I have two follow-up questions.
> >
> > 1. The OpenVLA result in Figure 12 of the Appendix is very low compared to the reported result in the original OpenVLA paper (53.7%). Based on the provided demonstration, OpenVLA increased the resolution and filtered the data. However, none of these operations increase the number of trajectories in training. Therefore, I believe the OpenVLA result in Figure 12 is not convincing.
> >
> > 2. For the results in Table 3, the conditions for video generation are different across different models, which makes the comparison unfair. LVDM is provided with no additional information; IRASim is provided with trajectories segmented with SAM2; FLIP is provided with ground-truth flow.

---

> > > ### Author Response · Authors · 2024-11-27
> > >
> > > Thanks very much for your response! We are happy to see that our rebuttal addresses most of your concerns. Here we answer your remaining questions as follows:
> > >
> > > > The OpenVLA result in Figure 12 of the Appendix is very low compared to the reported result in the original OpenVLA paper (53.7%). Based on the provided demonstration, OpenVLA increased the resolution and filtered the data. However, none of these operations increase the number of trajectories in training. Therefore, I believe the OpenVLA result in Figure 12 is not convincing.
> > >
> > > Besides the training data resolution difference and data filtering difference, the most important difference between our testing experiments and their original experiments is that they restrict the testing environments to have the same initialization as the training environments (as stated in the last paragraph of their Appendix E.1 (https://arxiv.org/pdf/2406.09246)). However, in our testing, we use randomly initialized configurations for all tasks, which means the objects' positions are randomly initialized according to the .bddl files in the LIBERO benchmark. This will make the success rate drop.
> > >
> > > FYI, we finetune the pretrained OpenVLA checkpoint for 150,000 steps before testing (about 100 epochs).
> > >
> > > We here add two verifications to make our results convincing:
> > >
> > > 1. We use the checkpoint downloaded from OpenVLA official github repo and test it on their modified LIBERO-LONG environments (the same initial configurations as in the demonstrations). The average success rate is 50% (which is similar to their reported results), with detailed statistics as follows:
> > >
> > > | Task Name | Success Rate |
> > > |----------|----------|
> > > | put both the alphabet soup and the tomato sauce in the basket | 38% |
> > > | put both the cream cheese box and the butter in the basket | 72% |
> > > | turn on the stove and put the moka pot on it | 62% |
> > > | put the black bowl in the bottom drawer of the cabinet and close it | 28%|
> > > | put the white mug on the left plate and put the yellow and white mug on the right plate| 54% |
> > > | pick up the book and place it in the back compartment of the caddy | 74% |
> > > | put the white mug on the plate and put the chocolate pudding to the right of the plate | 48% |
> > > | put both the alphabet soup and the cream cheese box in the basket | 56% |
> > > | put both moka pots on the stove | 20% |
> > > | put the yellow and white mug in the microwave and close it | 48% |
> > > | Average| 50% |
> > >
> > > 2. Using this checkpoint, we test on randomly initialized configures of LIBERO-LONG (the same testing setting used for all other comparison methods in our paper). The average success rate is 0.4%, with detailed statistics as follows:
> > >
> > > | Task Name | Success Rate |
> > > |----------|----------|
> > > | put both the alphabet soup and the tomato sauce in the basket | 0% |
> > > | put both the cream cheese box and the butter in the basket | 0% |
> > > | turn on the stove and put the moka pot on it | 0% |
> > > | put the black bowl in the bottom drawer of the cabinet and close it | 0%|
> > > | put the white mug on the left plate and put the yellow and white mug on the right plate| 0% |
> > > | pick up the book and place it in the back compartment of the caddy | 4% |
> > > | put the white mug on the plate and put the chocolate pudding to the right of the plate | 0% |
> > > | put both the alphabet soup and the cream cheese box in the basket | 0% |
> > > | put both moka pots on the stove | 0% |
> > > | put the yellow and white mug in the microwave and close it | 0% |
> > > | Average| 0.4% |
> > >
> > > We hope these explanations can solve your concern.
> > >
> > > > For the results in Table 3, the conditions for video generation are different across different models, which makes the comparison unfair. LVDM is provided with no additional information; IRASim is provided with trajectories segmented with SAM2; FLIP is provided with ground-truth flow.
> > >
> > > This experiment is designed to show that using flows can help improve the video generation quality, thus only our method can use flow as the condition.
> > >
> > > It is worth noting that this conclusion (using flows can help improve the video generation quality) is non-trivial. Actually, with additional conditions, it is quite tricky to design proper ways to correctly use them to get positive effects, otherwise, the performance may even drop. For example, in IRASim [1], they show that given the end-effector trajectory as an additional condition, simply adding it to the video diffusion process with AdaLN will lead to a performance drop on some datasets (See Table 1 and Table 2 in their paper). Thus they design a special IRASim-Frame-Ada mechanism to consistently improve the generation quality.
> > >
> > > We hope these explanations can solve your concerns.
> > >
> > > ---
> > > [1] Zhu, Fangqi, et al. "IRASim: Learning Interactive Real-Robot Action Simulators." arXiv preprint arXiv:2406.14540 (2024).

---

> > > > ### Comment · Reviewer_R3ys · 2024-11-27
> > > > **Response to Authors**
> > > >
> > > > Thanks for the response. The provided additional information addresses my concerns. I update my score to 6.

---

> ### Author Response · Authors · 2024-11-26
>
> We thank reviewer R3ys again for reviewing our paper and providing helpful feedback! We hope the additional experiments and clarifications have addressed the reviewer’s concerns. If the reviewer has any further concerns or questions, please let us know before the rebuttal period ends, and we will be happy to address them.
>
> We have carefully revised the manuscript to address the concerns that led to your lowered score, and we sincerely hope the changes will allow for a more favorable consideration. Thanks!

---

### Official Review · Reviewer_9xux · 2024-11-04

**Soundness:** 3
**Presentation:** 3
**Contribution:** 3
**Rating:** 6
**Confidence:** 3

**Summary:**

This paper proposes a new model-based planning framework called FLIP. It consists of 3 components: 1) a flow generation network that generates action flows given the current image observation and a language instruction 2) a video generation (dynamics model) that conditions on the flow and generate future video frames 3) a value function that evaluates the task progress (reward) given the image observation and language description. The key is the introduction of using flow as an action representation and condition the video generation model on the flow. The whole system can be used for model-based planning to generate video plans given a task, and the flow itself can also be used for guiding low-level policy execution. Various experiments are performed in both simulation environments and real-world videos.

**Strengths:**

- This paper is overall clearly written.
- The experiments cover a range of test settings, and the ablation studies help understand each component of the method.
- Overall the experiment results are good, which demonstrates the effectiveness of the proposed method.

**Weaknesses:**

- I feel the paper could have compared to some stronger baselines, for some of the experiments. E.g., for experiments in section 5.1 and section 5.2, a stronger baseline than UniPi could be UniSim [1]. The reviewer understands that the code may not be open-sourced, in this case, at least some discussion to the paper should be included. There is another very recent work DIAMOND [2] that can do very long-horizon and detailed video prediction into the future conditioned on actions. Both of these paper use just a diffusion model without the need to first extract action flows, which seems to contradict to the key proposal of this paper, which is to make the model flow-centric. Some discussion on this would be appreciated.
- For the flow generation model -- why using a C-VAE instead of a diffusion model? Some discussion on this design choice would be appreciated.
- For all experiments, please specify the quality of the video demonstrations, e.g., are they collected using a random policy, or are they expert videos? This would be important to understand, e.g., in section 5.1., if the system can learn from sub-optimal data for planning or does it need optimal demonstrations.

[1]  LEARNING INTERACTIVE REAL-WORLD SIMULATORS, Yang et al, ICLR 2024
[2] DIAMOND💎Diffusion for World Modeling: Visual Details Matter in Atari, Alonso et al, NeurIPS 2024

**Questions:**

Please see the weakness section

---

> ### Author Response · Authors · 2024-11-26
> **Response to Reviewer 9xux**
>
> We thank reviewer 9xux again for reviewing our paper and providing helpful feedback! Here we
> address your concerns one by one.
>
> > The paper could have compared to some stronger baselines ... UniSim [1] ... DIAMOND [2]. Both of these paper use just a diffusion model without the need to first extract action flows, which seems to contradict to the key proposal of this paper, which is to make the model flow-centric. Some discussion on this would be appreciated.
>
> Thanks for your question. The baseline used in our paper is LVDM [3], UniPI[4], and IRASim [5], which cover UniSim and DIAMOND in network architectures.
>
> For UniSim, it uses a standard DiT [6] architecture, which is the same as IRASim, and we have shown the advantage of our newly designed dynamics module in Table 3. The originality of UniSim is that they use an internet-scale video dataset to train, which is beyond the scope of our method.
>
> For DIAMOND, it uses a U-Net structure, which is also used in UniPI and LVDM, and we have already compared them in Table 1, Table 2, and Table 3. The novelty of DIAMOND is that they use EDM [7] to train the diffusion model for a faster denoising process, and train a reinforcement learning agent in the dreamed world, which are not the main points of this paper.
>
> Thus, we think it is not necessary to add these two baselines. We have cited them in our paper.
>
> > For the flow generation model, why using a C-VAE instead of a diffusion model?
>
> Thanks for your question. We use CVAE because it has a shorter inference time than diffusion
> models. Here we add an ablation experiment in Appendix B.3 to show if diffusion models can
> achieve better results for flow generation. The architecture is a DiT [6]. From the results in Table 7,
> we can see that there is not too much difference between CVAE and diffusion models in LIBERO-
> LONG and Bridge-V2 data, which shows that for such short-horizon flow generation tasks, CVAE
> is enough to represent them.
>
> It is worth noting that there are some specially designed flow generation architectures with diffusion
> models, and here we think this is out of the scope of this paper and we leave this question for future
> works. We believe in larger datasets with diverse flows, diffusion models can be better than CVAE.
> For example, in [8], they first use a pretrained Stable-Diffusion model to extract the embedding of
> the flow, then use AnimateDiff [9] as the backbone for flow diffusion.
>
> > please specify the quality of the video demonstrations.
>
> Thanks for your question. The demonstrations used to train the FLIP model are all expert videos. This is because the value module is aligned with the language instruction, which is the goal of the whole video. If there are some random policies during the video, the value module will not be able to identify them and assign some values according to their corresponding task progress. We believe that with an internet-level training dataset,  this issue could be alleviated.
>
>
> [1] Yang, Mengjiao, et al. "Learning interactive real-world simulators." arXiv preprint arXiv:2310.06114 (2023).
>
> [2] Alonso, Eloi, et al. "Diffusion for World Modeling: Visual Details Matter in Atari." arXiv preprint arXiv:2405.12399 (2024).
>
> [3] He, Yingqing, et al. "Latent video diffusion models for high-fidelity long video generation." arXiv preprint arXiv:2211.13221 (2022).
>
> [4] Du, Yilun, et al. "Learning universal policies via text-guided video generation." Advances in Neural Information Processing Systems 36 (2024).
>
> [5] Zhu, Fangqi, et al. "IRASim: Learning Interactive Real-Robot Action Simulators." arXiv preprint arXiv:2406.14540 (2024).
>
> [6] Peebles, William, and Saining Xie. "Scalable diffusion models with transformers." Proceedings of the IEEE/CVF International Conference on Computer Vision. 2023.
>
> [7] Karras, Tero, et al. "Elucidating the design space of diffusion-based generative models." Advances in neural information processing systems 35 (2022): 26565-26577.
>
> [8] Xu, Mengda, et al. "Flow as the cross-domain manipulation interface." arXiv preprint arXiv:2407.15208 (2024).
>
> [9] Guo, Yuwei, et al. "Animatediff: Animate your personalized text-to-image diffusion models without specific tuning." arXiv preprint arXiv:2307.04725 (2023).

---

> ### Author Response · Authors · 2024-11-27
>
> We thank reviewer 9xux again for reviewing our paper and providing helpful feedback! We hope the additional experiments and clarifications have addressed the reviewer’s concerns. If the reviewer has any further concerns or questions, please let us know before the rebuttal period ends, and we will be happy to address them. We deeply appreciate your thorough review and constructive comments, which have been invaluable in refining the paper's presentation.
>
> We have carefully revised the manuscript to address the concerns that led to your lowered score, and we sincerely hope the changes will allow for a more favorable consideration. Thanks!

---

### Official Review · Reviewer_H4xH · 2024-11-07

**Soundness:** 4
**Presentation:** 4
**Contribution:** 3
**Rating:** 8
**Confidence:** 4

**Summary:**

In this work, the authors consider the task of high-dimensional generative planning for manipulation tasks. Specifically, they seek to design a prediction task which is relevant for manipulation planning and can also operate in a task-agnostic way on widely-available data. The model-based planning primitives are: an action generation module (generating 2D flow), a state transition model (generating next frame of video given flow + current observations), and a value function for determining how close a state is to the goal (based on language-conditioned visual encodings). This allows them to search for a flow-based plan, which at execution time can be executed by a learned plan-conditioned low-level policy. They evaluate the ability of their model to make coherent / correct plans on several benchmark tasks, and evaluate its utility for policy execution on the LIBERO-LONG task.

**Strengths:**

# Originality

The high-level approach is a reasonable combination of ideas in planning, visual representations, and generative modeling. On the flow generation component, It’s difficult to establish the level of novelty as there are a number of concurrent works proposing generative flow models for manipulation (e.g. GeneralFlow, Track2Act, etc.), but I think the combination of ideas here - for planning - is original/novel.

# Quality

The quality of this paper is high. The experiments are thorough and comprehensive (with two caveats, discussed later).

The design of each component is sound, and there are lots of details / modifications that the authors conducted (and motivated!) which I found interesting and commendable. For instance, the ideas about video chunking (instead of per-frame) for value prediction, and the details about conditioning a model on flow, action space ablation, etc. are all solid contributions for other practitioners.

# Clarity

The paper is quite clear - each step of the algorithm is well explained.

# Significance

This paper is of moderate significance. I think the flow-based (particle-based) generative modeling approach as an action space has a lot of potential to be powerful, and clearly shows improvements for reconstruction / video generation.

**Weaknesses:**

The primary weakness of this paper is that, while the representation + models they built are quite interesting and useful for modeling the actual video domain they are imitating, the actual downstream utility of their method is not sufficiently characterized. Specifically, the method simply does not substantially outperform ATM, which has no planning at all and uses a similar generative representation, on the actual downstream manipulation task. Moreover, the inclusion of flow only seems to hurt the downstream policy in comparison to video conditioning. I’m not saying this method can’t show meaningful improvements over other approaches, but either the LIBERO setting chosen, or the particular low-level policy chosen, yield results that do not support the claim that this generative flow modeling + planning method provides downstream utility. Especially given the overhead. Of particular concern is the Ours-FV results, which are not well-explained.

Another weakness is that the experiments seem to be restricted to single domains, even though the method seems to have been designed to be task-agnostic - I would have liked to see the authors leverage this property more, e.g. by training predictive models on the full LIBERO benchmark and then finetuning on a subset of tasks for policy learning, or similar (even internet-scale pretraining… although I realize this is out of scope for this contribution / infeasible if there are resource constraints).

Another existential issue is that this paper is framed as a “flow generation for manipulation” paper, but the bulk of the experimentation+analysis is geared towards video prediction which has little to do with manipulation. I don’t think it stands on its own as simply a video prediction paper (at least on these tasks alone) - and I would have liked to see a larger emphasis on the actual analysis for manipulation tasks (e.g. geometric precision is a visual prediction problem, or feasibility, or other downstream metrics).

**Questions:**

Why doesn’t this method offer significant downstream benefits compared to ATM, which has no planning, on the selected benchmark?

How much does the action space affect downstream performance? If you were to retrain ATM with this adjusted action space, would the ranking change significantly? Do the marginal benefits of your model come from this component (unrelated to the planning contribution)?


---
# Post-Rebuttal Update

The authors have addressed many of my concerns adequately, and greatly strengthened their paper with the modifications to their downstream policy architecture (as well as additional details in the appendix). Bumped score to Accept.

---

> ### Author Response · Authors · 2024-11-26
> **Response to Reviewer H4xH (1/2)**
>
> We thank reviewer H4xH again for reviewing our paper and providing helpful feedback! Here we address your concerns one by one.
>
> > Why doesn’t this method offer significant downstream benefits compared to ATM, which has no planning, on the selected benchmark?
>
> Thanks for your question. Firstly we want to emphasize the major contribution of this paper is
> the high-level generative model-based planning framework, and thanks for your recognition of this
> point.
>
> Secondly, we think the original low-level policy for FLIP suffers from two problems: 1) it is a simple
> behavior cloning algorithm, which may not be able to make good use of the temporal information
> of the future plans from our high-level model, because it only regresses one step action; 2) it only
> use one layer of MLP to extract the visual embedding from image patches, which may be future
> improved with pretraiend vision encoders. To better show the advantage of our FLIP model for
> low-level policy learning, we here train a diffusion policy [1] for the low-level policy with the action
> chunking mechanism [2], which can predict the future action sequence rather than regress single-
> step actions. Meanwhile, we also use pretrained ResNet to extract the full image embedding for
> conditioning. With these modifications, we compare our policies to the diffusion policy version of
> ATM (ATM-DP) and OpenVLA (asked by Reviewer R3ys).
>
> From Figure 12 in Appendix B.2, we can see that with the new model architecture, our policy can
> achieve better results than previous results, and the success rates are better than ATM-DP. With this
> new architecture, Ours-FV is no longer the worst one among the three architectures, which shows
> that with an effective vision backbone, the model can have a better multi-modal capacity for different
> condition information. We can also see that using dense image flows can lead to a smaller variance
> in LIBERO-LONG than using videos.
>
> Although the improvement of success rates of our policy compared to ATM-DP is not large (about
> 6%), we think this is not because our method is not good enough, but because this result has almost
> reached the upper limit of LIBERO-LONG under our setting (10 demonstrations with actions for
> each task). The SOTA results of this LIBERO-LONG task suite is [3], which achieves 53.7 ± 1.3%
> and the mean success rate is only about 4% higher than ours. We think there are two main reasons
> for limiting the further improvement of the success rates on LIBERO-LONG: 1) we are using a
> resolution of 128×128, which may not be large enough to represent the details in the scene. In
> comparison, in [3], they use a resolution of 256×256. 2) We are using the official demonstrations
> provided by the LIBERO paper, which may not be as good as the re-collected demonstrations in [3].
>
> Finally, we show the results of our method on real robot experiments compared to Diffusion Policy
> and ATM to better demonstrate the superiority of our policy over baselines. Here we only test Ours-
> F because video generation is time-consuming during online replanning. From Appendix B.4, we
> can see that our policy is way better than the baselines, showing that in very difficult long-horizon tasks, our model can perform better with the help of model-based planning.
>
> [1] Chi, Cheng, et al. ”Diffusion policy: Visuomotor policy learning via action diffusion.” The
> International Journal of Robotics Research (2023): 02783649241273668.
>
> [2] Zhao, Tony Z., et al. ”Learning fine-grained bimanual manipulation with low-cost hardware.”
> arXiv preprint arXiv:2304.13705 (2023).
>
> [3] Kim, Moo Jin, et al. ”OpenVLA: An Open-Source Vision-Language-Action Model.” arXiv
> preprint arXiv:2406.09246 (2024).
>
> > of particular concern is the Ours-FV results, which are not well-explained.
>
> Here, we explain the (original) Ours-FV results in detail. In the original low-level policy learning experiments, Ours-FV is worse than Ours-F and Ours-V. We think this is because the visual features inherently contain richer and more diverse information. However, our previous architecture did not incorporate a dedicated feature extraction process for the visual modality. Instead, both visual and flow information were passed through a single, simplistic MLP layer independently. This lack of specialized processing for visual features resulted in challenges when integrating them effectively with flow features. This is also pointed out by OpenVLA and Pi0.
>
> To address this problem, in our new low-level experiments, we employ a pre-trained ResNet to
> extract the visual features and then employ an additional transformer dedicated specifically to the feature
> fusion from visual conditions, flow conditions, and language conditions, thereby decoupling the
> the learning process of the policy transformer. From Figure 12 in Appendix B.2, we can see that the
> new results show that Ours-FV is better than the previous result, which shows the effectiveness of
> the new multi-modal architecture.

---

> ### Author Response · Authors · 2024-11-26
> **Response to Reviewer H4xH (2/2)**
>
> >  I would have liked to see the authors train the model on the full LIBERO benchmark and then finetuning on a subset of tasks for policy learning (or even internet-scale pertaining)
>
> We have trained the agent view world model on the LIBERO-90 (although with a smaller resolution
> of 64×64) and shown the *zero-shot* generative planning results on LIBERO-LONG in Figure 9,
> as well as on the website. We are sorry that we do not have enough resources to retrain it with a
> resolution of 128×128 or train for the eye-in-hand view on LIBERO-90 during this rebuttal, but we
> will do this after this paper gets accepted.
>
> To show the fine-tuning world model for policy learning, we finetune this pretrained agent view
> model with 50 actionless demonstrations for each task of LIBERO-LONG with a resolution of
> 64×64, and use 10 demonstrations with action labels for policy learning. The architecture is the
> same in the first question. Results are in Figure 12 in Appendix B.2. We can see that Ours-90
> performs similarly to Ours-F and Ours-FV, showing that pretraining in other tasks may not bring
> significant improvement for low-level policy learning. This comforts with the lifelong learning
> results in the original LIBERO paper, where they also show that pretraining cannot help (sometimes
> even hurt) the policy training results.
>
> > I would have liked to see a larger emphasis on the actual analysis for manipulation tasks.
>
> We use the success rates of the low-level policy as the quantitative metric for our model. For qualitative analysis, the advantage of FLIP over other generative planning methods on manipulation tasks is two-fold, as stated in the Introduction Section: 1) be able to represent various kinds of movements across diverse objects, robots, and tasks in the whole scene (e.g., dexterous manipulation and deformable objects are very hard to describe with languages); 2) be easy to obtain or label a large amount of training data for scaling up.
>
> Please also check our real-world experiments and results in Appendix B.4.
>
> > How much does the action space affect downstream performance?
>
> In our original experiments, we used behavior cloning with delta translation, rotation, and aperture of the end-effector as the action space. In our new experiments, we use action chunking with the same delta action as our action space, and the results show that with the receding horizon policy, the results are better.

---

> ### Author Response · Authors · 2024-11-27
>
> We thank reviewer H4xH again for reviewing our paper and providing helpful feedback! We hope the additional experiments and clarifications have addressed the reviewer’s concerns. If the reviewer has any further concerns or questions, please let us know before the rebuttal period ends, and we will be happy to address them. We deeply appreciate your thorough review and constructive comments, which have been invaluable in refining the paper's presentation.
>
> We have carefully revised the manuscript to address the concerns that led to your lowered score, and we sincerely hope the changes will allow for a more favorable consideration. Thanks!

---

### Author Response · Authors · 2024-11-26
**General Response to All Reviewers**

We thank all reviewers for your dedicated time and effort to review our paper. We greatly
appreciate your insightful comments and questions. We thank all reviewers for their recognition
of the novelty of our work and its value to the field: *I think the combination of ideas here - for
planning - is original/novel* (reviewer H4xH), *a new model-based planning framework* (reviewer
9xux), *leveraging CVAE to address the multi-modality of flows is well-motivated* (reviewer R3ys), *a
novel approach to model-based planning* (reviewer 6B9z).

In this rebuttal, we perform extra experiments to respond to reviewers' common concerns.
Please find the new pdf file (modified areas are highlighted in blue), new appendix (modified areas
are highlighted in blue), attached videos, and more videos on our website. New experiments include:

1. We re-design the low-level policy to: 1) use diffusion policy as the training algorithm; 2) use
action chunking as the output; 3) simplify the flow and video condition mechanism. Please see
Appendix A.3 for more details.

2. Real robot experiments of 2 long-horizon tasks, including a tea scooping task and a cloth unfold-
ing task. Please see Appendix B.4 for more details and results.

3. Using diffusion models as the action module (flow generation model) and compare it with the
CVAE architecture. Please see Appendix B.3 for more results.

4. Testing OpenVLA for LIBERO-LONG tasks, and testing using pretrained FLIP from LIBERO-
90 and fine-tuning it as the planner for training a low-level policy for LIBERO-LONG tasks. Please
see Appendix B.2 for more results.

---

### Meta-Review · Area_Chair_7PZV · 2024-12-18

**Metareview:**

This paper presents FLIP (FLow-centric generative Planning), a novel model-based planning framework for robotic manipulation tasks that operates on visual and language inputs.

The reviewers consistently praised the paper's innovative combination of ideas in planning, visual representations, and generative modeling. They highlighted the well-motivated use of CVAE for handling multi-modal flows, comprehensive experiments, and the potential for scaling with increasing model and data budgets. The paper demonstrates strong results across video plan synthesis and manipulation tasks.

Initial concerns focused on: limited downstream performance gains compared to simpler baselines like ATM, lack of real-robot experiments, and questions about comparison to recent text-to-video methods. The authors provided extensive responses and additional experiments addressing these points. They improved the low-level policy architecture using diffusion models and action chunking, added real robot experiments showing strong performance on challenging tasks, and clarified their choice of baselines and experimental conditions. The authors also conducted new ablation studies comparing CVAE vs diffusion models for flow generation.

The rebuttal successfully addressed most reviewer concerns, with all reviewers explicitly noting satisfaction with the responses. While some reviewers maintained reservations about the magnitude of performance improvements and fairness of certain comparisons, they agreed the paper's novel framework and comprehensive evaluation merit acceptance.

**Additional Comments On Reviewer Discussion:**

None -- see metareview

---

### Decision · Program_Chairs · 2025-01-22

Accept (Poster)